# The anomalously thundery month of June 1925 in southwest Spain: description and synoptic analysis

**Francisco Javier Acero**[1], **Manuel Antón**[1], **Alejandro Jesús Pérez Aparicio**[1,2], **Nieves Bravo-Paredes**[1], **Víctor Manuel Sánchez Carrasco**[1], **María Cruz Gallego**[1], **José Agustín García**[1], **Marcelino Núñez**[3], **Irene Tovar**[3], **Javier Vaquero-Martínez**[4], **and José Manuel Vaquero**[1]

[1]Departamento de Física, Universidad de Extremadura, Badajoz, Spain
[2]Earth Remote Sensing Laboratory (EaRSLab) and Center for Sci-Tech Research in EArth sysTem and Energy (CREATE), Instituto de Investigação e Formação Avançada (IIFA), Universidade de Évora, 7000-671 Évora, Portugal
[3]Agencia Estatal de Meteorología (AEMET), Badajoz, Spain
[4]Departamento de Didáctica de las Ciencias Experimentales y de las Matemáticas, Universidad de Extremadura, Cáceres, Spain

**Correspondence:** José Manuel Vaquero (jvaquero@unex.es)

**Abstract.** In a routine search for meteorological events with a great impact on society in the Extremadura region (southwest interior of the Iberian Peninsula) using newspapers, the month of June 1925 was detected as exceptional due to the large number of thunderstorms associated with significant losses of human lives and material resources. This extraordinary month underwent a detailed examination from various complementary perspectives. Firstly, we reconstructed the history of the events, considering the most impacted locations and the resulting damage. Periodical publications, especially the widely circulated *Extremadura* newspaper in 1925, were pivotal in this regard. Secondly, we scrutinized monthly meteorological variables (precipitation, temperature, and cloudiness) using the lengthiest-available data series from the Iberian Peninsula. This aimed to underscore the exceptional characteristics of June 1925. Lastly, we analyzed the synoptic situation of the thunderstorm events by employing National Oceanic and Atmospheric Administration/Cooperative Institute for Research in Environmental Sciences/Department of Energy (NOAA/CIRES/DOE) 20th Century Reanalysis V3 (20CR) data. This approach allowed us to comprehend, from a synoptic perspective, the exceptional nature of this month. Thereby, a combination of a negative North Atlantic Oscillation (NAO) situation, elevated convective available potential energy (CAPE) values, large-scale lifting, and abundant precipitable water availability in the region was revealed.

## 1 Introduction

Thunderstorms are essential phenomena for understanding the climate system (Markson, 2007; Rycroft et al., 2008). In addition to their scientific interest, thunderstorms have important consequences for our society since they produce a huge variety of dangers and problems such as heavy rain, lightning, large hail, and tornadoes (Holle, 2016; Antonescu et al., 2017; Prein and Holland, 2018). The scattered nature of all these phenomena has made their study and prediction difficult until a few decades ago, when large databases became available for the scientific community (see, for example, Dotzek et al., 2009, and Taszarek et al., 2021).

The area affected most by thunderstorms on the Iberian Peninsula is located in the northeast, especially in the mountainous regions of the Pyrenees (north Catalonia and Aragon) and the Iberian system (south Aragon). A climatology of stormy days and electrical discharge was recently published by Núñez Mora et al. (2019). In the scientific literature, several exceptional thunderstorm events in these areas in the northeast of the Iberian Peninsula can be found. For example, several authors have studied thunderstorms that have pro-

duced exceptional episodes of hail, such as the events that occurred in July 2001 (Tudurí et al., 2003), in September 2004 (Ceperuelo et al., 2006), or in June 2006 (Montanyà et al., 2009). In addition, other exceptional cases have been studied, such as the severe thunderstorm on 4 October 2007 that affected the island of Mallorca (Ramis et al., 2009) or the convective system that affected Catalonia on 21 March 2012, which produced a tornado (Bech et al., 2015). In all these cases, convective activity was very intense, although both the patterns in the general circulation of the atmosphere and the different local aspects can be very different. Climatological studies on thunderstorms on the rest of the Iberian Peninsula are scarcer. For example, Ezcurra et al. (2008) studied the rain characteristics of thunderstorms in the north of the Iberian Peninsula during the 5-year period of 1992–1996. The establishment of lightning detection networks allowed scientists to carry out interesting studies for periods of around 10 years (Rivas Soriano et al., 2005; Santos et al., 2013). In addition, other studies have analyzed the impact of thunderstorms on social and economic aspects, such as wildfires (García-Ortega et al., 2011).

In this context, we discovered a notable set of news items about thunderstorms in the Spanish historical press during the month of June 1925. These journalistic reports strongly caught our attention since the geographical area where they occurred, the interior of the southwest of the Iberian Peninsula, is one of the regions of the Iberian Peninsula with the fewest days of thunderstorms per year, and the consequences described by journalists were exceptional. Therefore, the objectives of this article are (i) to make a detailed description of the detrimental effects on lives, goods, and infrastructure of that extremely stormy month of June 1925 in southwest Spain from news collected in newspapers; (ii) to carry out an evaluation of the observed meteorological data (precipitation, temperature, and cloudiness) even though these events occurred almost a century ago; and (iii) to analyze the synoptic situation that caused these exceptional thunderstorms.

## 2 Datasets and methodology

### 2.1 Historical sources

The historical press of the region of Extremadura (in the southwest of the Iberian Peninsula) has been consulted to obtain information about the meteorological events. In particular, we analyzed the newspaper *Extremadura*, which led us to discover the unusual period of thunderstorms affecting this region that occurred in 1925. The newspaper *Extremadura* was the most important newspaper in the region at that time, together with the newspaper *Hoy*, which appeared later in 1933. Subsequently, the virtual newspaper library of the Spanish government (https://www.prensahistorica.mcu.es, last access: 7 November 2024) has also been consulted for the period between 15 May and 15 July 1925. The main Ex-

tremadura newspapers consulted in this library are *La Montaña* and *Correo de la Mañana*. In addition, one national newspaper, *La Correspondencia de España*, has been analyzed. We found 11 reports of thunderstorm events in Extremadura in the newspaper *Extremadura*, 9 in the newspaper *La Montaña*, 9 in the newspaper *Correo de la Mañana*, and 2 in the newspaper *Correspondencia de España*. Some characteristic examples of the news reports found can be seen in Fig. 1, and some basic information about them is listed in Table 1. From all of them, a database has been created describing each event, its location, the date of the event, and the publication of the news, as well as information on the impacts of the event such as economic impacts, human losses, and injured people.

### 2.2 Meteorological data and reanalysis

The Spanish Meteorological Agency (*Agencia Estatal de Meteorología*, AEMET) provided the records for the time series construction of the three meteorological variables analyzed in this work: precipitation ($P$), temperature ($T$), and cloudiness ($N$).

The relationship between the thunderstorm events and rainfall has been studied based on 64 accumulated monthly precipitation series homogenized by AEMET (Luna et al., 2012). These rainfall time series cover 158 years, from 1851 to 2008. Moreover, daily rainfall time series for seven locations over the Extremadura region were used to analyze the short-term variability in precipitation in this region during June 1925.

With the goal of checking the relationship between the thunderstorm events and temperature during June 1925, daily temperature records have been analyzed in this work using 20 long and reliable Spanish time series (Brunet et al., 2006).

The cloudiness observed in June 1925 over Spain was analyzed in this work by means of the parameter of cloudiness at 39 stations covering 146 years, from 1865 to 2010. Thus, the parameter of cloudiness (PC) used in our work to characterize the cloudiness is defined (in percentage) as

$$PC = 50 + 50 \cdot ((O - C)/N), \tag{1}$$

where $O$ and $C$ are the number of overcast and cloudless days, respectively, and $N$ is the number of days in a given period (month, season, year). We have used the data provided by Sanchez-Lorenzo et al. (2012), who inferred monthly series of the variable given by Eq. (1) from the number of cloudless and overcast days recorded every month at 39 Spanish stations from 1865 to 2010. For that, those authors recovered monthly series of cloudless and overcast days from different volumes of the publication entitled *Resumen de las observaciones meteorológicas efectuadas en la Península*, edited by AEMET, from 1865 to 1960. Since 1961, daily cloud cover data have been provided in digital format by AEMET, and, consequently, the parameter of cloudiness (Eq. 1) was derived from monthly frequencies of

**Figure 1.** News clippings from the newspapers *Extremadura*, *Correo de la Mañana*, and *La Montaña* (courtesy of the central library of the University of Extremadura).

**Table 1.** Date, newspaper name, title, [translated title], and a summary of the news that is reproduced in Fig. 1 (from left to right).

| Date and newspaper name | Title | Summary |
|---|---|---|
| 15 June 1925 *La Montaña* | *La tormenta de esta tarde ha sido de primera clase y de gran aparato "escénico"* [This afternoon's thunderstorm was first class and had great "scenic" effects] | There was heavy rain and deafening thunder in the Cáceres area. It was similar to the thunderstorm that occurred on 7 June. |
| 15 June 1925 *La Montaña* | *Furiosa tormenta. Un joven muere ahogado, sin que aparezca su cadáver* [Raging thunderstorm. A young man drowns, but his body has not been found] | Raging thunderstorm in Zarza de Granadilla. A shepherd drowns while crossing the Aldevara stream. The body is not found, despite the efforts of law enforcement and family members. |
| 11 June 1925 *La Montaña* | *La tormenta del miércoles* [Wednesday's thunderstorm] | A violent thunderstorm. The worst damage was in Malpartida de Cáceres, with three people injured by lightning. |
| 9 June 1925 *Correo de la mañana* | *Horrorosa tormenta* [Horrible thunderstorm] | Formidable thunderstorm in Segura de León: streets and houses are flooded, roads and highways are impassable, and there is a great impact on agricultural activities. |
| 11 June 1925 *Correo de la mañana* | *De Zafra. Dos ahogados* [From Zafra. Two drowned] | A huge thunderstorm caused the Peñaranda stream to rise. Two people drowned at Don Adrián's flour mill, where they were caught by a strong flood. |

cloudless and overcast days in order to cover the 1961–2010 period. Capel Molina (1981) established that a day is defined as cloudless if the mean cloud cover from several daily observations is lower than 20 %, while is defined as overcast if this mean is higher than 80 %. Thus, if the cloud cover is recorded in oktas, the thresholds could be less than 1.5 for cloudless days and greater than 6.5 for overcast days.

Figure 2 shows the distribution of $P$, $T$, and $N$ stations on the Iberian Peninsula (circles and dots). In addition, this plot also displays the location of seven $P$ stations with daily data (inverted triangles) located in the Extremadura region.

Additionally, we used the latest version (version 3) of the National Oceanic and Atmospheric Administration/Cooperative Institute for Research in Environmental Sciences/De-

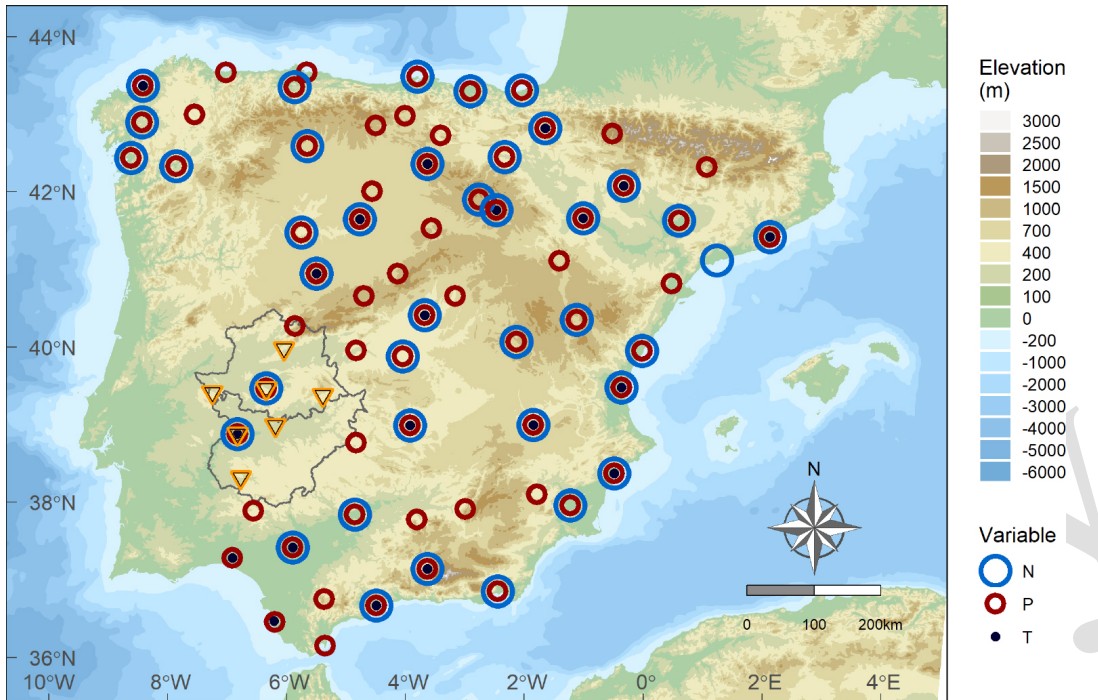

**Figure 2.** Map of the Iberian Peninsula with the borders of the region of Extremadura and its two provinces. The observatories are marked with blue circles (monthly cloudiness data, $N$), red circles (monthly precipitation data, $P$), or black dots (daily temperature data, $T$). Moreover, observatories with daily precipitation data in the region of Extremadura are shown with inverted yellow triangles.

partment of Energy (NOAA/CIRES/DOE) 20th Century Reanalysis (20CR) data (provided by the NOAA Physical Sciences Laboratory, Boulder, Colorado, USA, from their website at https://psl.noaa.gov, last access: 7 November 2024) (Compo et al., 2011; Slivinski et al., 2019). This has been made possible by the latest data assimilation systems and several sets of historical meteorological observations. This particular dataset is well-suited for the intended analysis, as it offers a continuous three-dimensional depiction of numerous meteorological variables dating back to 1836, encompassing a significantly longer period compared to the standard National Centers for Environmental Protection/National Center for Atmospheric Research (NCEP/NCAR; since 1948) or European Centre for Medium-Range Weather Forecasts (ECMWF; since 1958) reanalysis datasets. In particular, 20CR uses an ensemble filter data assimilation method, thus providing a direct estimation of the most likely state of the global atmosphere (for each 3 h period). Moreover, there is also an estimation of the uncertainties in that reanalysis. Evaluating the historical performance of the 20CR reanalysis is not a simple task since it is impossible to make comparisons with other reanalyses, and it can only be done by comparing with independent observations (Slivinski et al., 2021). Some comparison exercises carried out have been satisfactory. In particular, in our study area, the 20CR results were satisfactory for the extreme precipitation event of autumn 1876 in the Guadiana River basin (Trigo et al., 2014).

The upper level (250 hPa) information from the 20CR reanalysis will also be used in this work. It should be noted that it was derived primarily by statistical methods for the period examined and is not the result of a standard reanalysis. This means that it has a much higher level of uncertainty than the sea level pressure fields or the upper-level information for periods where radiosonde information is available.

## 3 Historical description of the stormy month of June 1925

This episode of thunderstorms that occurred in June 1925 had a great impact throughout Extremadura. Figure 3 shows the positions and names of the numerous towns and villages located in the north, center, and mainly south of Extremadura, where different kinds of damage caused by the thunderstorms was reported. Extremadura exhibits diverse orography, significantly influencing its hydrological patterns. The region has mountainous terrain, such as the Sierra de Gata and Sierra de San Pedro (in the north and west, respectively), with mountains above 1000 m in height, which act as natural barriers to moist air masses from the Atlantic. Conversely, the plains in the south, like La Serena or La Campiña, provide fertile ground for agriculture and livestock. Moreover, there are several important rivers in Extremadura. The main rivers are the Guadiana and the Tajo, which flow from east to west. Other smaller rivers are the Alagón, Tiétar, Zújar, Sa-

lor, Ardila, and Guadiato. These rivers play a crucial role in the region's climate, as they serve as conduits for moisture and influence local weather patterns. The region's orography influences the air mass movement, especially in the northern mountainous areas, where orographic lift leads to higher precipitation levels. Of course, the rivers contribute to the region's humidity levels, enhancing cloud formation and precipitation.

The regional Extremadura newspapers included extensive information on the thunderstorms of June 1925 and their impact on the region. An overview of the thunderstorms and their impacts according to the newspaper reports is presented below.

The largest city where reports of thunderstorms have been found is Cáceres. This is the most important city in the province of Cáceres, one of the two provinces of the region of Extremadura. According to reports in the newspapers *La Montaña* and *Extremadura*, there was a heavy thunderstorm in Cáceres on 7 June, another one on 10 June, a third one around 14–15, June and a fourth one on 19 June. During three of them (7, 10, and 14–15 June) there was flooding of streets and houses. Furthermore, the thunderstorm on 7 June lasted for 2 h, during which there were several lightning strikes, one of which caused a widespread power blackout in the city. On the other hand, on 10 June the thunderstorm lasted only 10 min, but it was of great intensity, with torrential rain and huge hailstones that severely damaged the countryside. The center of these two thunderstorms was the area of the city of Cáceres, with no rainfall in the surrounding area.

In other places, deaths were reported during some thunderstorms, such as occurred in the Zafra, Villalba, Bienvenida, and La Lapa zones on 10 June, where a total of four people died: two of them drowned due to the enormous flooding of the Peñaranda riverbank, and the other two were struck by lightning in the hut where they were sheltering from the thunderstorm, according to the newspapers *Extremadura* and *Correo de la Mañana*. Another death occurred in Zarza de Granadilla when a man was swept away by the current while trying to ford a stream on 10 June, as reported in the newspaper *La Montaña*. The death of a child who drowned when she was swept away by a stream in the thunderstorm in Berlanga is also to be regretted, according to the news item of 22 June in the newspaper *Correspondencia de España*, where it is also stated that lightning killed three people in Llerena. The newspaper *Extremadura* reports that in the village of Montemolín, there were 15 consecutive days of thunderstorms, killing a man when he was struck by lightning. The same newspaper also reports that another person died from the same cause in the thunderstorm that occurred in Montánchez on 8 June. However, the event with the highest number of deaths was the thunderstorm on 18 June in Higuera de Vargas, according to the newspaper *Correo de la Mañana*, in which five people died when they were struck by lightning while sheltering in a hut.

As well as the fatalities, several people were injured, and animals were killed. For example, in that same hut in Higuera de Vargas, apart from the death of those five people, four people were injured, and eight pigs that were in the vicinity died. Moreover, according to the reports from the newspaper *La Montaña*, two people were also injured in the thunderstorm on 10 June in Cáceres. Two people suffered burns when they were struck by lightning in Malpartida de Cáceres, and three donkeys were killed by the lightning, according to the same newspaper. In addition, many animals drowned in different locations.

Another one of the most frequent impacts of the thunderstorms was the floods that occurred in many places. According to the news reported in the newspaper *Correo de la Mañana*, in Segura de León a strong thunderstorm around 7–8 June caused the flooding of a multitude of houses and streets. In addition, the strong flow of water caused the watercourses to break in several places, sweeping away animals, devastating the fields, and leaving the trunks of holm oaks bare due to the impact of the stones carried by the current. The same newspaper reports that further north, in Ribera del Fresno, there were also major floods due to a thunderstorm on 16 June. The most insignificant stream was transformed into a mighty river, and the streets carried so much water that it was impossible to cross them. In some houses the water reached a height of 1 m, collapsing walls and sweeping away everything in its path. A few days later, in the same area, the newspaper *Extremadura* reported a major thunderstorm on 25 June in the village of Hinojosa del Valle, during which the whole village was flooded, and several houses were destroyed. In addition, it is reported that a stream overflowed its banks in Jerez de los Caballeros due to another thunderstorm on 21 June. The overflowing of the Bodión River, the Peñara riverbank, and the Guadiana River in the thunderstorm on 10 June in the Zafra area mentioned above must not be forgotten.

It is worth mentioning the damage caused to infrastructure by the intense thunderstorms. There were collapsed bridges, such as the one over the Víar River during the thunderstorm on 6 June in the area of Montemolín, according to the newspaper *Correo de la Mañana*. Another bridge over the Tagus River fell due to the thunderstorm on 7 June in the area of Santiago del Carbajo, according to the newspaper *La Montaña*. In addition, it is reported that traffic between Santiago del Carbajo and a nearby village called Herrera de Alcántara was interrupted. The collapse of houses and walls was also very common in many towns during these thunderstorms, as occurred in Segura de León, Cáceres, Malpartida de Cáceres, Hinojosa del Valle, and Ribera del Fresno.

Crop and field damage was extensive in many of the locations where thunderstorms developed, leading to a major economic impact due to the region's dependence on agriculture at that time. For example, a thunderstorm in Alconera on 7 June destroyed crops and trees, leaving only the subsoil in many places, according to the newspaper *Correo*

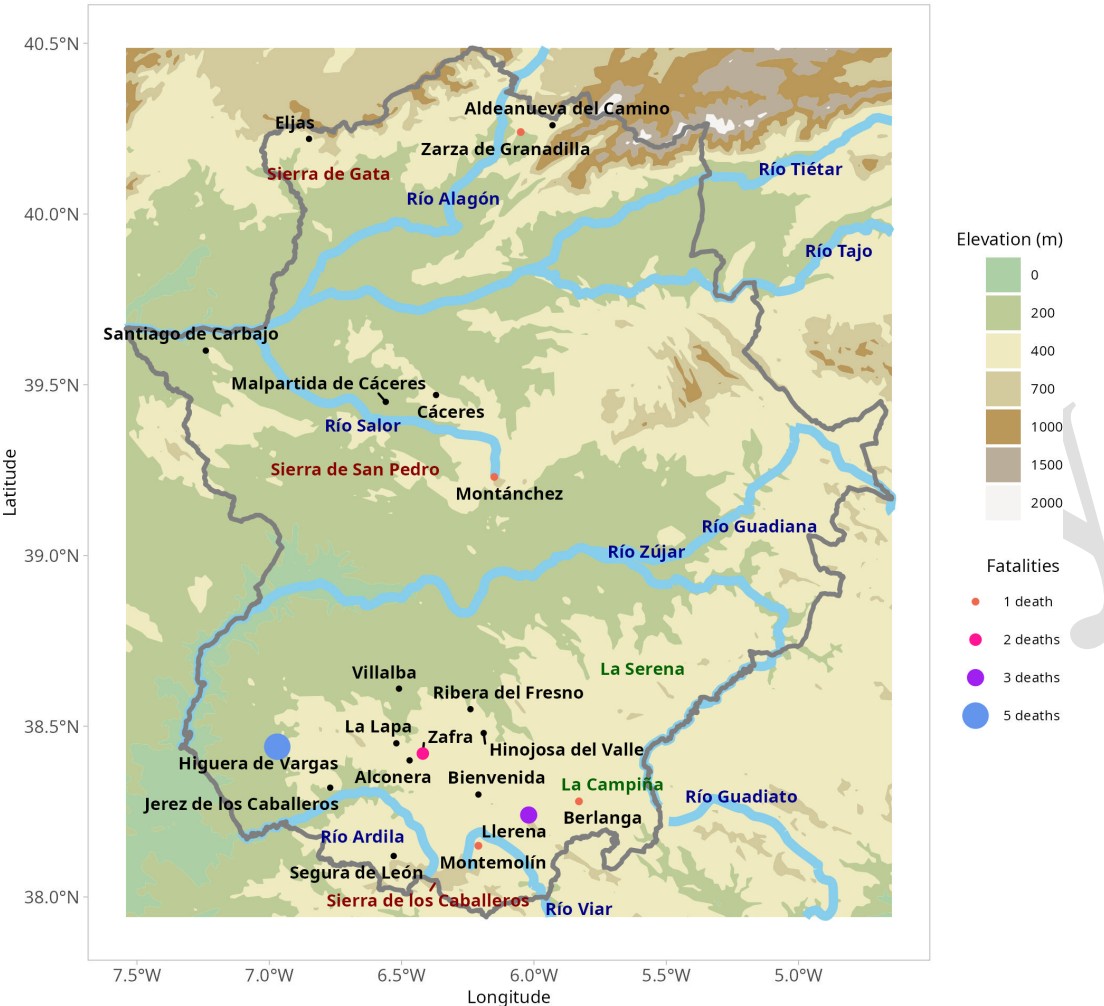

**Figure 3.** Geographical distribution of the Extremadura locations affected by the storms that occurred in June 1925, according to the documentary sources consulted in this work. Color shows the number of deaths directly related to the thunderstorm events, which were extracted from the documentary sources (black dots mean no deaths were reported).

*de la Mañana*. Something similar happened on 10 June in Aldeanueva del Camino and on 18 June in Eljas, according to reports from the newspaper *Extremadura*, where the water and hail caused considerable damage to the orchards.

## 4  Assessing the observed instrumental data

As this episode of thunderstorms in June 1925 led to strong impacts throughout Extremadura, it is necessary to analyze the behavior of rainfall in this month. For this purpose, daily rainfall data from seven locations over Extremadura were used. Figure 4 shows daily rainfall in June 1925 for these observatories. The local character of precipitation during thunderstorms is revealed. Most observatories recorded precipitation between 2 and 6 June, with Cornalvo (in the center of the study area) being the one with the highest values. During the rest of the month, thunderstorms and precipitation are more

isolated, appearing at some observatories, while there was no rain at others. Thunderstorms with rainfall higher than $20\,\mathrm{mm\,d^{-1}}$ were recorded on 2–8, 13, 16, and 18 June.

In order to analyze whether the accumulated rainfall in June 1925 was remarkable, Fig. 5 shows the ranking of that month compared to the remaining 157 June months for the time series of each observatory in peninsular Spain. The eight observatories marked in red represent the places where June 1925 was the wettest or the second-wettest June and are located in the southwest. In this same area, for most of the observatories, the rainfall recorded in June 1925 is among the 10 rainiest months of June for the whole time period. On the contrary, there are four observatories in the northwest showing that June 1925 was one of the driest months of June.

For the three meteorological variables analyzed in this work (precipitation, temperature, and cloudiness), the standardized anomalies between June 1925 and the average of

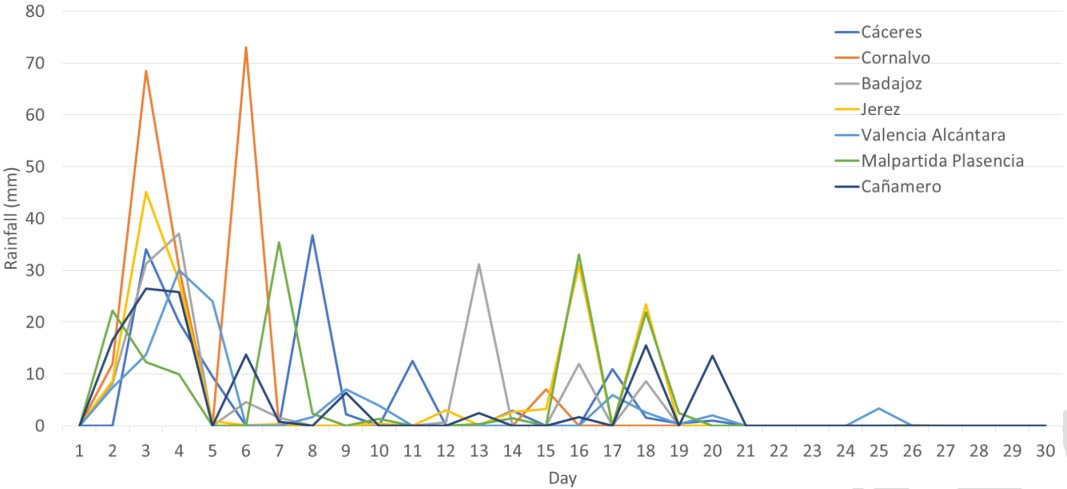

**Figure 4.** Daily rainfall recorded at seven observatories located in Extremadura in the month of June 1925.

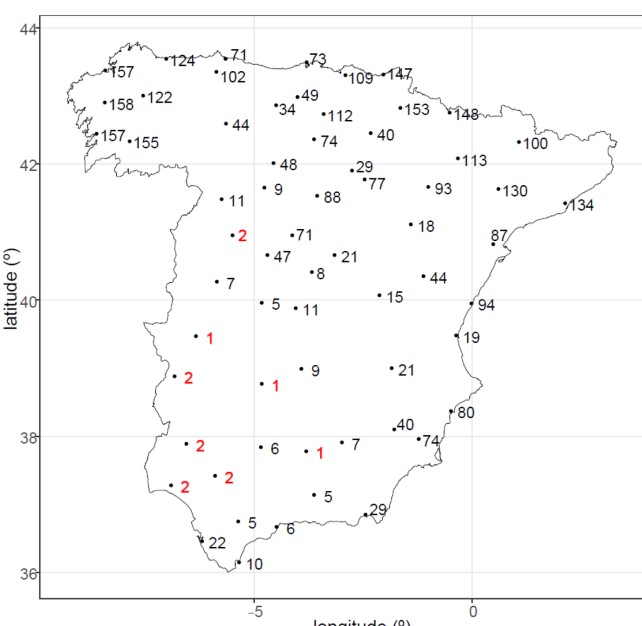

**Figure 5.** Spatial distribution of the rankings representing the accumulated rainfall in June 1925 among the other June months in the 158 years (1851 to 2008) that make up the complete time series for each observatory. Red numbers represent the observatories where June 1925 is the wettest or the second-wettest June.

June of the corresponding variable have been estimated as follows:

$$Y = \frac{X_{\text{June1925}} - \overline{X}_{\text{June}}}{\text{SD}(X_{\text{June}})}, \tag{2}$$

with $X_{\text{June1925}}$ as the value for the variable in June 1925 and $\overline{X}_{\text{June}}$ and $\text{SD}(X_{\text{June}})$ as the respective mean and standard deviation of the variable for the month of June for the whole

time series. In this section, variables such as rainfall, temperature, and cloudiness are analyzed.

Figure 6a shows the rainfall anomalies for 64 time series located over peninsular Spain. Note that in order to allow a better interpretation of the spatial behavior of the results, the anomalies were spatially interpolated by a kriging procedure. The highest anomalies are located over the southwest of Spain, with the study area showing anomalies over 3; i.e., in June 1925 it rained between 3 and 4 times more than normal in an average June. For these observatories, June 1925 shows the highest accumulation of rainfall in 158 years. The rainfall anomalies decrease towards the north and northeast of Spain.

When studying the relationship between temperature and thunderstorm events, it can be expected that the temperature will be lower than usual in a month as rainy as the one that occurred in the study area. Figure 6 (central panel) shows the monthly temperature anomalies for our time series. Anomalies showing a colder-than-average June 1925 lie in the southwest, although they are weak. Similarly to the rainfall, the temperature anomalies decrease towards the northeast of Spain. Moreover, Fig. 6 (right panel) shows the spatial variability in the monthly cloudiness anomalies for June 1925 with respect to the average for the 1866–2010 period in Spain. A clear dependence on latitude can be seen, with negative cloudiness anomalies for all northern locations and positive anomalies for the central and southern sites. In addition, we see that the central and southwestern regions of Spain present the highest cloudiness anomalies. Several locations exhibit extremely high cloudiness values in June 1925 compared to all months of June between 1866 and 2010. For example, June 1925 had an absolute cloudiness record in Madrid, Cuenca, and Granada. It also had the second-highest values in Badajoz, Toledo, and Málaga.

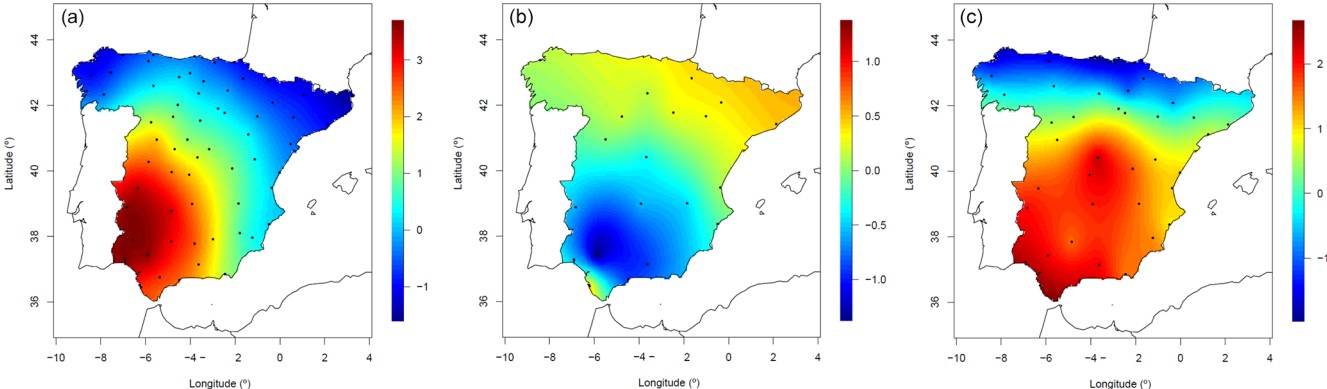

**Figure 6.** Rainfall (**a**), temperature (**b**), and cloudiness (**c**) anomalies for June 1925.

## 5    Synoptic analysis leading to the June 1925 events

In addition to the analysis of temperature, precipitation, and cloudiness series, the synoptic situation of each day of June 1925 is analyzed in order to understand the reason for the stormy events during the month. For this purpose, the 20CR reanalysis data were used to carry out the analysis. The wind vector (streamlines) and the geopotential height at 250 hPa for each day of June 1925 are plotted in Fig. 7. Jet streams are a core of strong westerly winds located in the upper levels of the troposphere. Therefore, the jet stream is easily identified in Fig. 7. In summer, the polar jet stream is weaker than in winter, and this favors a wavier flow. The polar jet stream in the first days of June reached $50 \, \mathrm{m \, s^{-1}}$, and the flow began to ripple (Fig. 7). The wave broke on 3 June, bringing on a cutoff low located over the southwest of the Iberian Peninsula. During the next few days, the polar jet stream continued to be wavy, and an anticyclone began to form poleward of the cutoff low. This situation can be assimilated to a blocking system (Barriopedro et al., 2010; Lupo, 2021).

The cutoff-low-pressure system was one of the prominent patterns during June 1925, and the corresponding convection increased precipitation that was locally very intense. This could also explain the increase in cloudiness and the lower-than-usual temperatures for the month of June in this region. Note that the persistent trough and cutoff-low patterns shown at 250 hPa and also at 500 hPa are compatible with a strong low-level southern flow (700 or 850 hPa) over the area of study, especially around the province of Badajoz, where there is usually a flow from the south and southwest at low levels. However, orographic reinforcement of precipitation is not typical in the south of the province of Badajoz, since the mountains, even if they were aligned perpendicular to the flow, are not high enough. This effect is well known upwind of the southern flow, in the Sierra de los Caballeros (the peak of Tentudía at 1104 m and the western summit of Los Bonales at 1053 m), but the locations affected by the storms in 1925 (Fig. 3) are all in the lee of the aforementioned flow.

In fact, the entire province of Badajoz, except for the southern mountains, can be considered geographically to be a large valley of the Guadiana River, open to the west-southwest. That is why this orographic forcing of precipitation does not occur here. Perhaps the specific orography in locations such as Jerez de los Caballeros, Higuera de Vargas, La Lapa, etc., could have had some influence not on the precipitation but on its channeling and could have generated some local effects such as flooding or overflows.

Synoptic pattern classifications are a useful analytical tool for understanding the weather of a region. We will use the synoptic pattern classification established by Font-Tullot (1983, 2000) to analyze the synoptic situation of each day of June 1925. Specifically, we will use the new pattern classification carried out by Santos et al. (2019), which updates and improves the well-known Font-Tullot classification for the Iberian peninsular region. This synoptic classification consists of 23 different patterns. Santos et al. (2019) used the ERA40 reanalyses to review the objective classification of Ribalaygua-Batalla and Borén-Iglesias (1995). Moreover, the subjective classification of Font-Tullot (1983) was retrieved in detail, proposing 23 synoptic patterns that are illustrated with situations from 23 specific dates from the 1970s to the 1980s.

The geopotential height at 500 hPa and the sea level pressure (SLP) are analyzed for each day in order to identify which pattern corresponds to each day. Table 2 shows the seven patterns identified for June 1925. Five different patterns are identified between 1 and 22 June, and all are associated with thunderstorms (except for pattern no. 16, not associated with thunderstorms, and no. 21, uncertain) by Santos et al. (2019). The most common patterns are no. 5 (Azores anticyclone and peninsular thermal depression), no. 18 (Ibero-African barometric trough), and no. 21 (barometric dam). Figure 8 shows an example of these three patterns, showing the SLP (blue contour lines) and the geopotential height at 500 hPa (colored shading). Pattern nos. 5, 18, and 21 are represented in Fig. 8 on the left (2 June), in the center (10 June), and on the right (18 June), respectively. Pattern

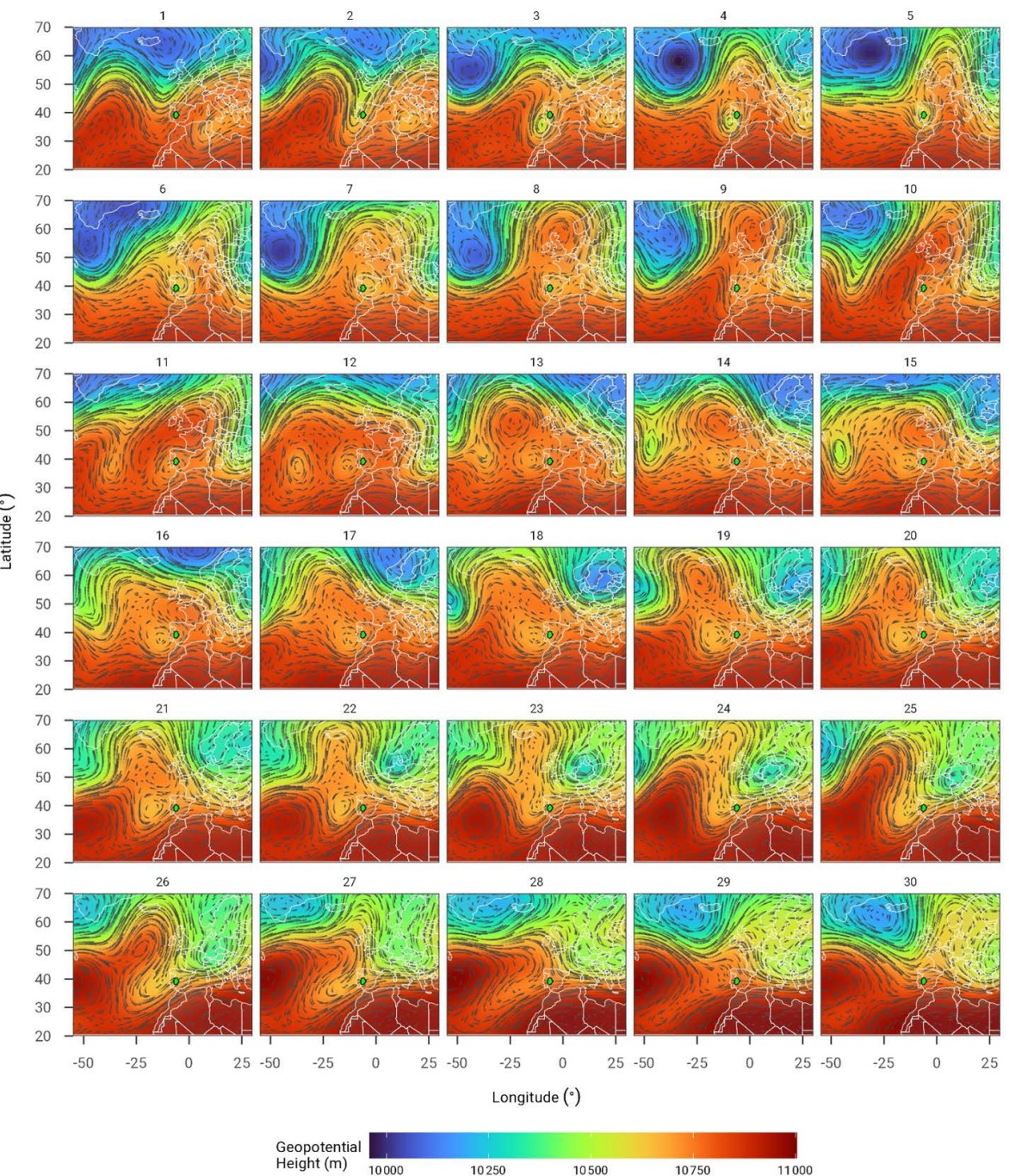

**Figure 7.** Wind vector (streamlines) and geopotential height at 250 hPa for each day of June 1925. The Extremadura region is shown in green.

no. 5 is associated with storms between May and September, being more frequent in July and August. In addition, pattern no. 18 is common in June and is associated with fair weather, although it could be associated with cutoff lows in southern Spain. Finally, pattern no. 21 is associated with fair weather with occasional storms, especially in the north of the Iberian Peninsula. Between 23 and 30 June 1925, the most common pattern was no. 10. This pattern is associated with cold and dry weather in southern Spain. As can be seen in Sect. 3 and Fig. 4, most of the stormy and rainy days occurred between 1 and 22 June. In fact, as discussed in Sect. 4 in relation to Fig. 4, thunderstorms with rainfall higher than $20\,\mathrm{mm\,d^{-1}}$ were recorded on 2–8, 13, 16, and 18 June. All these days except 8 June are associated with patterns that could be com-

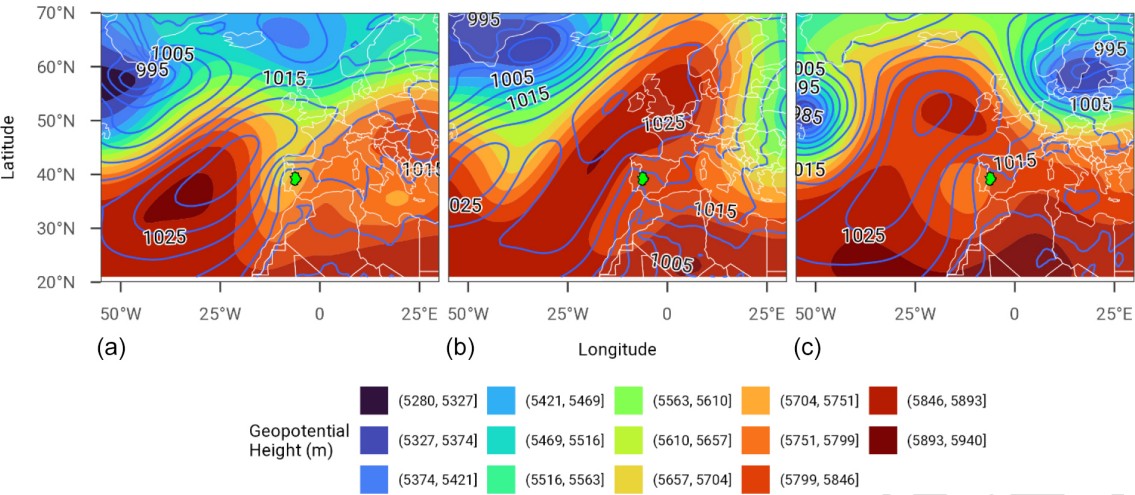

**Figure 8.** The synoptic situations of 2 June (**a**), 10 June (**b**), and 18 June (**c**) showing an example of pattern types no. 5, no. 18, and no. 21, respectively, according to the classification by Santos et al. (2019). Geopotential height at 500 hPa is represented by the colored shading and SLP by the blue contour lines CE1. The Extremadura region is shown in green.

patible with thunderstorms or rain (see the last column in Table 1). As evident in Sect. 3 and Fig. 4, most stormy and rainy days occurred from 1 to 22 June. Consequently, the synoptic analysis conducted in this time period aligns with the observations documented in the newspapers.

Lastly, we have generated synoptic charts of the main meteorological fields, as well as different composites of the monthly mean values and anomalies regarding the climatological period covered by the 20CR reanalysis. Following Doswell et al. (1996), thunderstorms and deep, moist convection require three ingredients: moisture, instability, and lift. Vertical wind shear is also required to allow storm organization (e.g., Markowski and Richardson, 2010). In the current article, precipitable water content (moisture), CAPE (instability), and omega ($\mathrm{d}p/\mathrm{d}t$, lifting) are analyzed.

A summary of our results is presented in Figs. 9 and 10. Figure 9 is made up of six panels. The top two panels show SLP, while the middle two panels depict convective available potential energy (CAPE), and the bottom two panels display total precipitable water. The panels on the right present the composite means of the variables for June 1925, while the panels on the left exhibit the composite anomaly.

The top panels in Fig. 9 show a typical negative North Atlantic Oscillation (NAO) situation, with low pressure west of the British Isles and negative SLP anomalies in the southwest of the Iberian Peninsula. The middle panels of Fig. 9 reveal that the west of the Iberian Peninsula had high CAPE values in the context of the Atlantic and Mediterranean region, with positive mean anomalies in the west of the Iberian Peninsula during June 1925 (the values shown correspond to the composite mean of the entire month). Finally, the bottom panels present high values of precipitable water in the entire atmosphere in the southwest of the Iberian Peninsula, with the highest values of the anomaly over the region of Ex-

tremadura. Note that these monthly anomalies are calculated from the composite mean value (the climatology time period selected for the calculation is 1981–2010). Therefore, the exceptional month of June 1925 in Extremadura was characterized by a combination of a negative NAO situation, high CAPE values, and precipitable water available in this area. In any case, note that Fig. 9 shows that the largest CAPE in Spain for June 1925 was not located precisely in southwestern Spain but was in northwestern Spain and northern Portugal. It seems that the 20CR reanalysis for such early times gives us significant patterns, although perhaps the exact location of the details is a little displaced.

The panels in Fig. 9 are complemented by panels in Fig. 10 that show the omega field at several pressure levels during June 1925 (left panels), as well as the omega field at the same pressure levels during all June months for the period of 1851–2014 except 1925 (right panels). Results in the left panels show a negative anomaly in omega at all pressure levels during June 1925 in the west of the Iberian Peninsula, up to a very high level far from the surface ($\sim 150\,\mathrm{hPa}$), where the lift seems to have disappeared. Moreover, while the sea level pressure anomaly is negative across much of Europe (Fig. 9), the west of the Iberian Peninsula stands out as a region with stronger-than-average large-scale lifting when looking at the omega field. This indicates that large-scale lifting could have been a relevant factor for the development of the thunderstorms in June 1925. Results in the right panels show that this negative anomaly is non-existent for June months for the full period of the reanalysis (1851–2014) except 1925. It indicates the exceptionality of June 1925, researched in this work and not included in the European Severe Weather Database (ESWD; Dotzek et al., 2009), which only includes four severe weather phenomena for the year 1925 in eastern Spain.

**Table 2.** Patterns identified in June 1925, according to the classification by Santos et al. (2019).

| Pattern | Brief description | Day of the month | Storm or rain |
|---|---|---|---|
| No. 5 | Azores anticyclone and peninsular thermal depression | 1–3, 6, 7, 28, 29 | Yes |
| No. 8 | Atlantic anticyclone and peninsular thermal depression | 4, 5 | Yes |
| No. 10 | Gulf of Genoa depression | 24–27 | No |
| No. 16 | British–Scandinavian anticyclone | 8, 9 | No |
| No. 18 | Ibero-African barometric trough | 10–13 | Yes |
| No. 20 | Summer peninsular cold depression | 23 | Yes |
| No. 21 | Barometric dam | 14–22 | Uncertain |

## 6 Conclusions

Thunderstorms are crucial for understanding the climate system and have significant societal implications due to their various hazards. The northeastern region of the Iberian Peninsula, particularly the mountainous areas of the Pyrenees and the Iberian system, is highly affected by thunderstorms. Studies have examined exceptional thunderstorm events in this region, including episodes of hail and severe thunderstorms. Climatological studies on storms on the Iberian Peninsula are limited but have explored rain characteristics and the impacts on social and economic aspects, such as wildfires. A notable set of news reports from June 1925 in the interior southwest of the Iberian Peninsula drew our attention due to the region's infrequent storms and the exceptional consequences described by journalists. In this study, we have provided a detailed description of the detrimental effects during that stormy month. Moreover, we have evaluated instrumental data from almost a century ago and have analyzed the synoptic situation that caused these exceptional thunderstorms.

The thunderstorms that occurred in June 1925 had a significant impact throughout Extremadura, Spain. Numerous towns and villages in the north, center, and south of Extremadura reported various kinds of damage caused by the thunderstorms. The city of Cáceres experienced multiple storms in June, with flooding of streets and houses on 7, 10, and 14–15 June. The thunderstorms in Cáceres were characterized by heavy rain, lightning, and large hailstones, which caused power outages and severe damage to the countryside. Other areas such as Zafra, Villalba, Bienvenida, La Lapa, Zarza de Granadilla, and Berlanga also reported deaths and injuries from lightning strikes, flooding, and stream currents. Animals were affected as well, with several cases of dead animals due to lightning strikes or drowning. Flooding and overflowing of rivers and streams were widespread, leading to damaged houses, streets, and fields. Bridges, houses, and walls collapsed, and crops and orchards suffered extensive damage. The economic impact on agriculture was significant due to the destruction of crops and trees. These storms had a profound impact on the region, causing the loss of lives, injuries, infrastructure damage, and economic losses.

During the thunderstorms in June 1925 in Extremadura, the behavior of rainfall in the region was analyzed. Daily rainfall data from seven locations in Extremadura were examined, revealing the local nature of precipitation during thunderstorms. The highest values of precipitation were recorded between 2 and 6 June, with Cornalvo station experiencing the most significant rainfall. In the rest of the month, there were more isolated thunderstorms and varying precipitation patterns across the observatories. Several days, including 7, 8, 13, 16, and 18 June, had thunderstorms with rainfall exceeding $20\,\mathrm{mm}\,\mathrm{d}^{-1}$.

To determine whether the accumulated rainfall in June 1925 was exceptional compared to other June months, a ranking analysis was conducted. The eight observatories in the southwestern region of peninsular Spain marked in red in Fig. 5 had either the wettest or the second-wettest June on record in 1925. Most observatories in this area ranked among the top-10 rainiest June months throughout the entire dataset. In contrast, four observatories in the northwest indicated that June 1925 was one of the driest June months. We also examined standardized anomalies for precipitation, temperature, and cloudiness in June 1925 compared to the long-term averages (1850–2003). The rainfall anomalies were highest in the southwest, indicating that June 1925 had 3 to 4 times more rainfall than the average for June. The anomalies decreased towards the north and northeast of Spain. Temperature anomalies were lower than average in the rainy study area, with colder temperatures observed in the southwest. Cloudiness anomalies showed a clear dependence on latitude, with negative anomalies in northern locations and positive anomalies in central and southern regions. Central and southwestern Spain had the highest cloudiness anomalies, with several locations experiencing extremely high cloudiness compared to all other months of June from 1866 to 2010. Overall, June 1925 in Extremadura had significant rainfall, lower temperatures than usual, and increased cloudiness, particularly in the southwestern region.

We have analyzed the synoptic situation in June 1925 to understand the occurrence of stormy events during that month. The 20CR reanalysis data were used to examine the wind vector and geopotential height at 250 hPa for each day of June 1925. The presence of a polar jet stream and its wavi-

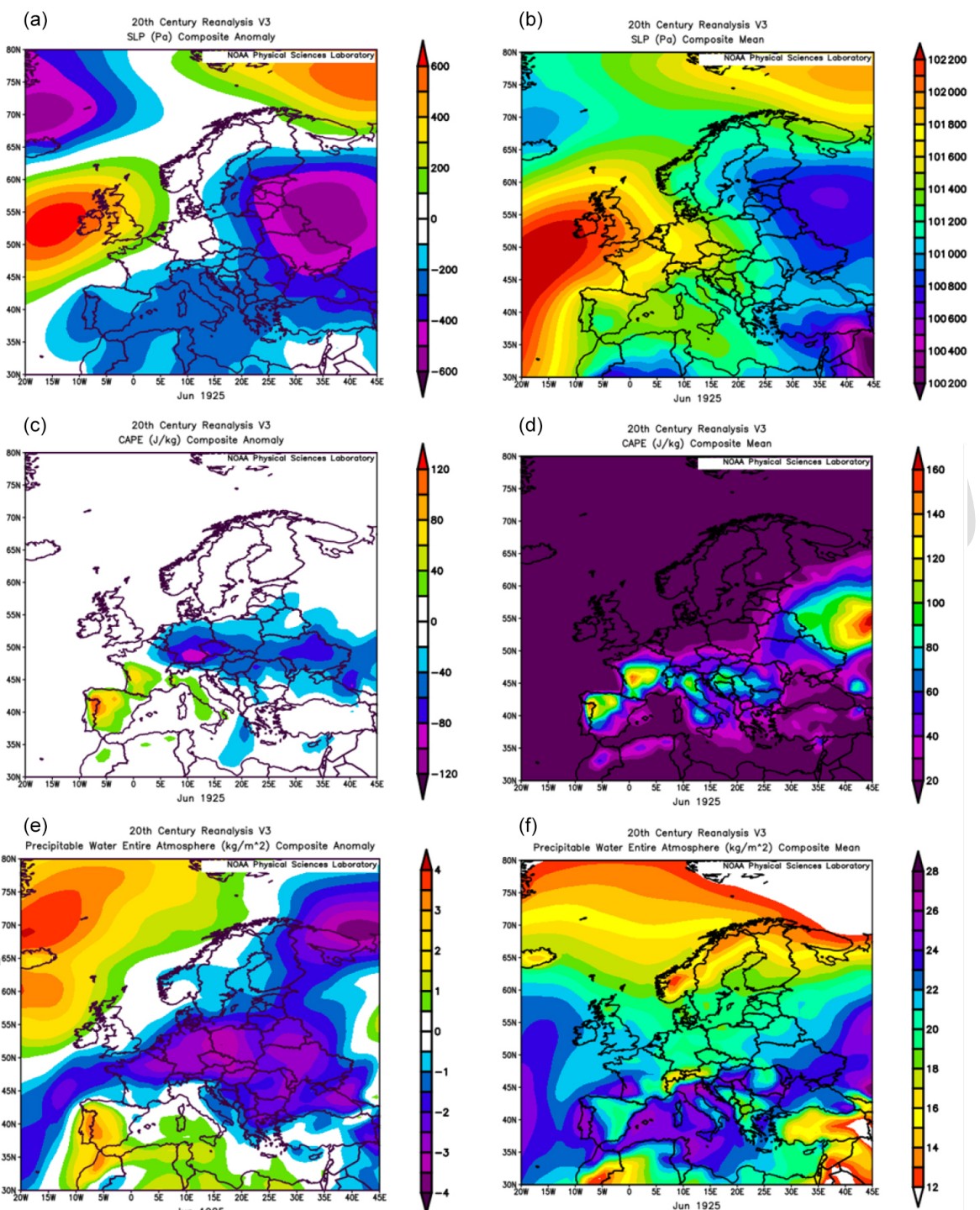

**Figure 9.** The composite mean **(b, d, f)** and composite anomaly **(a, c, e)** of SLP, CAPE, and the precipitable water entire atmosphere for June 1925 in the study area (top, middle, and bottom panels, respectively) from the 20CR reanalysis.

ness were observed, indicating a wavy flow pattern. The daily synoptic situations during this month show patterns associated with thunderstorms and rainfall on most days. Synoptic charts and composites of monthly meteorological fields for June 1925 were also generated. Our analysis suggests a nega-tive NAO situation, with low pressure west of the British Isles and negative SLP anomalies in the southwest of the Iberian Peninsula. Moreover, we have found high CAPE values in the west of the Iberian Peninsula, with positive mean anoma-lies during June 1925, and high values of precipitable water

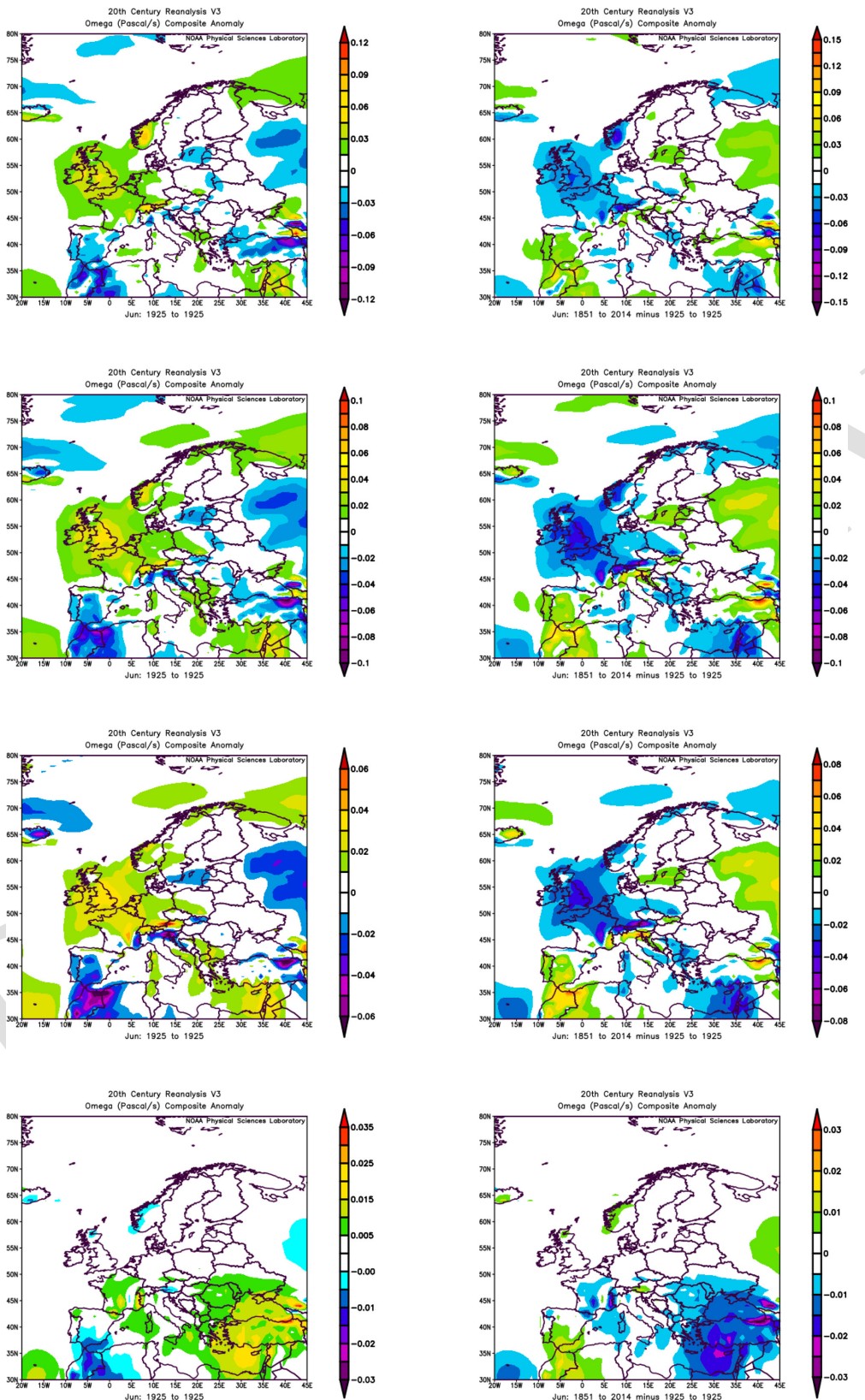

**Figure 10.** TSI The composite anomaly of omega ($\mathrm{d}p/\mathrm{d}t$) for June 1925 (left panels) and the composite anomaly of omega ($\mathrm{d}p/\mathrm{d}t$) for June months from 1851 to 2014 excluding 1925 (right panels) in the study area for pressure levels of 600, 500, 400, 200, and 150 hPa (from top to bottom) from the 20CR reanalysis.

in the southwest of the Iberian Peninsula, particularly in Extremadura. Overall, the exceptional month of June 1925 in the southwest of the Iberian Peninsula was characterized by a combination of a negative NAO situation, high CAPE values, large-scale lifting, and abundant available water in the region.

The analysis carried out in this article sheds light on the most extreme convective processes that can occur over the southwest of the Iberian Peninsula. The interest in these processes is enormous due to their catastrophic consequences.

*Data availability.* All raw datasets used in this study are public. Daily data of temperature and precipitation for Spain are available at https://www.aemet.es/es/serviciosclimaticos/cambio_climat/datos_diarios?w=2&w2=2 (last access: 7 November 2024) TS2. Cloud cover data for Spanish stations were provided by Sanchez-Lorenzo et al. (2012, https://doi.org/10.5194/cp-8-1199-2012). National Oceanic and Atmospheric Administration/Cooperative Institute for Research in Environmental Sciences/Department of Energy (NOAA/CIRES/DOE) 20th Century Reanalysis (20CR) data were provided by the NOAA Physical Sciences Laboratory, Boulder, Colorado, USA, from their website at https://psl.noaa.gov (last access: 7 November 2024) TS3.

*Author contributions.* JMV planned the research; NB-P, IT, and JMV extracted the information from the newspapers; FJA, MA, NB-P, MCG, JAG, MN, and JMV made the formal analysis of the data; FJA, MA, MCG, JAG, MN, IT, and JMV wrote the manuscript draft; and FJA, MA, AJPA, NB-P, VMSC, MCG, JAG, MN, IT, JV-M, and JMV reviewed and edited the article.

*Competing interests.* The contact author has declared that none of the authors has any competing interests.

ther geographical representation in this paper. While Copernicus Publications makes every effort to include appropriate place names, the final responsibility lies with the authors.

*Acknowledgements.* This research was supported by the Economy and Infrastructure Counseling of the regional government of Extremadura through project IB20080 and by the Department of Education, Science, and Vocational Training of the regional government of Extremadura through grant GR24049 (co-financed by the European Union). Alejandro Jesús Pérez Aparicio thanks the Universidad de Extremadura and Ministerio de Universidades of the Spanish Government for the award of a postdoctoral fellowship "Margarita Salas para la formación de jóvenes doctores" (MS-11).

*Financial support.* This research has been supported by the Junta de Extremadura (grant nos. IB20080 and GR24049) and the Ministerio de Universidades (postdoctoral fellowship "Margarita Salas" MS-11).

*Review statement.* This paper was edited by Vassiliki Kotroni and reviewed by two anonymous referees.

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

**Remarks from the language copy-editor**

CE1 Please note the small grammar adjustment here.

**Remarks from the typesetter**

TS1 Could you please provide a brief description of the necessary figure change? We will forward this request then to the handling editor of your paper for their approval. Thank you.

TS2 Please provide a reference list entry corresponding to this URL including creator/host of the data the URL leads to, a title for the reference, data repository, the URL and the year of the last access date.

TS3 Please see comment above about a necessary reference list entry for this URL as well. An example could be: 11NOAA Physical Sciences Laboratory: Reanalysis (20CR) data, https://psl.noaa.gov, last access: 7 November 2024." Please adapt this suggestion according to your preferences and provide a reference list entry similar in style for the aemet URL above. Many thanks.

TS4 Please provide date of last access.