# Peer review of "The anomalous thundery month of June 1925 in SW Spain"

_EGUsphere, 2023_

## Author Response (AR1)

**Authors reply in bold**

Referee #1

General Comments

This is an interesting study about a relevant historical month of active thunderstorms causing considerable damage. Both the period selected with the historical description of damages and the methodology used represent a valuable contribution, considering the difficulties of accessing the documentation of damages and the relative lack of observational datasets, compared to current standards. The manuscript is well suited for publication, but a number of clarifications and corrections should be made before further consideration, so I recommend a major review. Please find below specific comments including also suggestions for possible improvement.

**Thank you very much for your report. We also believe that this is a significant historical case deserving attention. While accessing information has indeed been challenging, we feel that we have managed to build a fairly comprehensive picture of the unusual meteorological events that took place in June 1925. We have addressed all your comments, clarifications, and corrections. We are truly grateful for all these insights, which will undoubtedly enhance the quality of our manuscript.**

Specific Comments

Page 1, line 20. Suggest: thunderstorms -> thunderstorm

**Done.**

Page 1, line 23 (and page 14, line 261). Please correct (two changes): available water -> total water vapour available [or precipitable water available]

**Done.**

Page 1, line 27. Reference Holle 2016: not found. Is it Holle 2006 (listed but not cited)? Please check and correct.

**We are sorry. The correct reference is Holle (2016): Holle, R. L.: A summary of recent national-scale lightning fatality studies, Wea. Clim. Soc., 8, 35–42. 2016. DOI: 10.1175/WCAS-D-15-0032.1**

**We have changed 2006 to 2016 in the reference list.**

Page 3, Figure 1. Showing some examples of news in the newspapers to illustrate the study is a good idea. However, the language of this journal is English so the information (title, subtitle and perhaps the whole text) should be translated into this language to be understood by the readers. I recommend presenting the information in a more systematic way, for example, as a table with different columns showing for each row (news): date, newspaper name, title, subtitle/text, for each of the 5 news presented.

**We think that is a good idea. We have included a new Table including date, newspaper name, title, subtitle/text or summary for each of the five-news presented in Figure 1. Thus, it is a perfect complement to Figure 1, especially for those readers who cannot understand the Spanish language. The new Table 1 is:**

**Table 1. Date, newspaper name, title, and a summary of the news that are reproduced in Figure 1 (from left to right).**

| Date and newspaper name | Tittle | Summary |
|---|---|---|
| 15/06/1925 La Montaña | La tormenta de esta tarde ha sido de primera clase y de gran aparato "escénico" [This afternoon's thunderstorm was first class and had great "scenic" effects] | There was heavy rain and deafening thunder in the Cáceres area. It was similar to the thunderstorm that occurred on June 7. |
| 15/06/1925 La Montaña | Furiosa tormenta. Un joven muere ahogado, sin que aparezca su cadaver [Raging storm. A young man drowns, but his body is still unavailable] | Raging storm in Zarza de Granadilla. A shepherd drowns while crossing the "Aldevara" stream. The body is not found, despite the efforts of law enforcement and family members. |
| 11/06/1925 La Montaña | La tormenta del miércoles [Wednesday's thunderstorm] | A violent storm. The worst damage was in Malpartida de Cáceres, with three people injured by lightning. |
| 09/06/1925 Correo de la mañana | Horrorosa tormenta [Horrible thunderstorm] | Formidable thunderstorm in Segura de León: streets and houses are flooded, roads and highways are impassable, and there is a great impact on agricultural activities. |
| 11/06/1925 Correo de la mañana | De Zafra. Dos ahogados [From Zafra. Two drowned] | A huge storm caused the Peñaranda stream to rise. Two people drowned at Don Adrián's flour mill, where they were caught by a strong flood. |

Page 4. Use of 20th Century Reanalysis: authors (and readers) should be aware of the limitations of this dataset: as opposed to other reanalysis such as ERA5, please indicate on the text if this product is only based on surface reports, and, in that case, that this implies that the upper-level fields should be used with caution as they are derived statistically from surface patterns. They are not physically interpolated form radiosonde data, unavailable at that time. Therefore, part of the results (Figure 7, 8, etc.) should be taken as possible, plausible patterns, but are not comparable to a reanalysis based on surface and radiosonde or other upper air data (satellite, etc.) not available at that time.

**Thank you so much. We believe that this is another good suggestion to improve our manuscript. We have included new text discussing the limitations of this dataset (20CR v3). Additionally, we have explained the implications of this on our results shown in Figures 7, 8, and 9. Please, see our new version of the manuscript (last paragraph of section 2.2 and section 5).**

Page 5 and 6, section 3. If I understood correctly, most damage was caused by lightning, large hail and flooding, sometimes with large rainfall differences between nearby locations which seems to indicate well organized deep moist convection, probably multicell or perhaps supercell storms. However, authors do not mention damage caused by strong winds (straight line winds, tornadoes, microbursts, etc.) which could well occur with such convective storms – could you please confirm explicitly in the text if there are any damage reports that could be linked to damage caused by strong winds from convective origin? For example, on the fourth news fragment of Figure 1 ("Horrorosa Tormenta"), if I understood correctly, it indicates that a gate made of logs and wire was destroyed and the rests were found one km away; or in page 16, line 285 'walls collapsed' are mentioned – it is not clear to me if those damages were caused by flooding or perhaps by strong winds.

**Thank you so much. This comment is also very interesting. We have reviewed all the information obtained from newspapers to check in more detail what exactly is described about the wind. People were killed by lightning or drowned in floods. Journalists perhaps report more of this than other details. In the case of the fourth news item ("Horrorosa Tormenta") shown in Figure 1, the wind is not mentioned, and the damage seems to be caused exclusively by a lot of rain, hail and lightning (but not strong winds). Please, see new Figure 1.**

Page 7, Figure 3. Please complement the current figure with basic geographical information such as terrain height, main rivers, etc. One possibility is adding another panel with the same geographical domain as current Figure 3 but with this information, instead of merging everything in one single panel figure. As mentioned below the terrain height distribution (mountain heights and orientation, etc.) might be relevant to interpret possible orographic effects upon precipitation.

**We have changed Figure 3 attending to the reviewer comment, including terrain height and main rivers (Figure 3 in the new version of the manuscript). We have also added the following paragraph:**

**Extremadura exhibits a diverse orography, significantly influencing its hydrological patterns. The region has mountainous terrain, such as the Sierra de Gata and Sierra de San Pedro (in the north and west, respectively), with mountains above 1000 m height, which act as natural barriers to moist air masses from the Atlantic. Conversely, the plains in the south, like La Serena or La Campiña provide fertile ground for agriculture and livestock. Moreover, there are several important rivers in Extremadura. The main rivers are the Guadiana and the Tajo, which flow from east to west. Other smaller rivers are the Alagón, Tiétar, Zújar, Salor, Ardila and Guadiato. These rivers play a crucial role in the regions climate as they serve as conduits for moisture and influence local weather patterns. The region's orography influences the air mass movement, specially in the norther mountainous areas, where orographic lift leads to higher precipitation levels. Of course, the rivers contribute to the region's humidity levels, enhancing cloud formation and precipitation.**

Page 7, Figure 3 caption. Suggest: in 1925 -> in June 1925. Then: to the 1925 thunderstorm events -> to the thunderstorm events

**Done.**

Page 7, line 170. The dates listed do not match daily rainfall records shown in Figure 4 (for example during the 3 June, and 6 June, the 20 mm/day threshold is exceeded at Cornalvo and Jerez). Please check and correct. Or do you mean that on those days rainfall was not caused by thunderstorms? Please clarify.

**Thank you very much for this comment. We understand now that we have not explained ourselves well. We have rewritten that phrase. Here we want to highlight the dates on which rainfall exceeded the threshold of 20 mm per day in any of the available stations. In any case, the region of Extremadura is very large (41,635 km²) so the seven available stations do not cover the entire region and in some cases, they may not be as significant of the whole region as we would like them to be. Therefore, the case could occur that there was no precipitation greater than 20 mm/day in the seven stations while in some locality in the region this value was greatly exceeded due to a local storm. We can write new text explaining this in more detail.**

**In particular, note that the dates mentioned in the text did fit with those in Figure 4. It happened that in the text we had only mentioned the dates beyond 20 mm that are outside the period June 2-6, since the analysis made in the text was separated for the period June 2-6 and for the following days (where dates that exceed 20 mm are cited). In any case, we have rewritten the paragraph so that, in the new version of the manuscript, the paragraph contains all the dates exceeding 20 mm. Please, see the new version of the manuscript.**

Page 7, equation 1 and line 184. Could you please use the standard symbol for average (a small line over X instead of underlining X)?

**Yes, of course. It was a problem in the pdf version. It was corrected and now it is right in our documents.**

Page 9, Figure 5 caption. Please indicate here the period covering the 158 years considered (despite the information may be given in the text I would add it here as well).

**Done. The new Figure 5 caption is: "Figure 5: Spatial distribution of the rankings representing the accumulated rainfall in the month of June 1925 among the other June months in the 158 years (1851 to 2008) that make up the complete time series for each observatory. Red numbers represent the observatories where June 1925 is the first or the second wettest June."**

Page 9, line 202-203. Suggest: temperature anomalies -> monthly temperature anomalies (similarly in line 205 for cloudiness). I think it is important to emphasize you're considering monthly anomalies.

**Yes, we agree. We have made this change. It is important to emphasize the monthly nature of anomalies.**

Page 10, line 214 and elsewhere in the text. Please use hPa instead of mb, as recommended by WMO (2008): "The unit "pascal" is the principal SI derived unit for the pressure quantity. The unit and symbol "bar" is a unit outside the SI system; in every document where it is used, this unit (bar) should be defined in relation to the SI. Its continued use is not encouraged."

**Yes, we agree. We have used hPa instead of mb.**

Page 10, line 215: atmosphere -> troposphere

**Done.**

Page 11, Figure 7. Please add units to the x and y axis titles (degrees). Please check units of geopotential height, are they m or gpm?

**Thank you. We have added units to the x and y axis titles (degrees) and we have checked units of geopotential height (they are m), as you can see in the attached figure (Figure 7 in the new version of the manuscript).**

Page 10 (comment on Figure 7) and page 12 (comment on Figure 8). The persistent trough and cut-off low pattern shown at 250 hPa and also at 500 hPa seems to be compatible with a strong low level southern flow (700 hPa or 850 hPa) over the area of study.

**We agree with this statement about Extremadura in general and especially about the province of Badajoz, where there is usually a flow from the south and southwest at low levels. Please, see the new version of our manuscript (section 5).**

If present, this could be an important factor as could transport southern warmer and moister air increasing atmospheric instability and, at the same time, causing cross barrier flow (due to mountain systems oriented west to east) which would increase vertical air speed and could favour orographic enhancement of precipitation (see for example Houze 2012 for an overview of this effect or case studies such as those described in Trapero et al 2013). I think this possibility could be briefly outlined on the text, pointing to the possibility of orographical effects enhancing heavy rainfall.

**We agree, in part, since this flow of warmer, more humid air usually occurs in these situations and increases atmospheric instability a little. However, we do not believe that the aforementioned orographic reinforcement of precipitation occurs in the south of the province of Badajoz, since the mountains, even if they were aligned perpendicular to the flow, are not high enough. This effect is well known upwind of the southern flow, in the Sierra de los Caballeros (the peak of Tentudía 1104 m and the western summit of Los Bonales 1053 m), but the locations affected by the storms in 1925 (figure 3) are all in the lee of the aforementioned flow.**

**The entire province of Badajoz, except for the southern mountains, can be considered geographically as a large valley of the Guadiana River, open to the west-southwest. That is why this orographic forcing of precipitation does not occur here. Perhaps the specific orography in locations such as Jerez de los Caballeros, Higuera de Vargas, La Lapa, etc., could have had some influence not on the precipitation but on its channeling and could have generated some local effects such as flooding or overflows.**

**This type of orographic forcing of precipitation, with flows from the south or southwest, does occur in areas of the province of Cáceres such as Las Villuercas (Pico Villuercas 1603 m) or in the regions of Jerte and La Vera where the elevations also reach higher heights. at 1500 m. In these regions mentioned, the average annual precipitation reaches much higher values than in the rest due mainly to its orientation perpendicular to this south-southwest flow.**

**Please, see the new version of our manuscript (section 5).**

Page 12, Figure 8. Could it be possible to plot the first and second row panels on a single one, i.e. by plotting for example the first one as shaded colours and the second one as contour line field, perhaps in a bit larger panel? This would allow to see better the relation between the two fields.

**We have changed Figure 8 according to the reviewer's suggestion and we agree that this way it is easier to interpret the information, so we have accepted the change in the new version of the manuscript (Figure 8).**

Page 12, line 231. Reference: Font -> Font-Tullot (listed but not cited) ?

**Thank you so much. We have made this change. Spanish authors usually sign with two surnames and that always generates some problems in Anglo-Saxon texts.**

Page 13, 1rst paragraph. Please look for an alternative to the term 'calm weather', I don't think it is precise enough for a scientific text.

**Thank you very much. We have looked an alternative to the term "calm weather". Santos et al. (2019) wrote "El tiempo es generalmente bueno" for both patterns 18 and 21. The English translation of "buen tiempo" is: "fine weather", "good weather" or "fair weather". We have change "calm weather" by "fair weather" in the new version of our manuscript.**

Page 13, Table 1.The list of days exceeding 20 mm/day (page 7, last line) seems to contradict the last column of Table 1. Please check.

**Thank you so much. We have checked the last column of Table 1 and incorporated new text in section 5. Please note that there is only one day that contradicts the last column of the table (day 8), while the other 4 days (7, 13, 16 and 18) do agree with the column. In fact, in the new version of the manuscript, as we have previously mentioned in another question from the reviewer, we have also mentioned June 2-6 as days that exceed 20 mm. Those days also agree with the last column of the table. So 9 matches with the last column of 10 days exceeding 20 mm is fine (note that the 20CR reanalysis should be used with caution for such early dates).**

References

Houze Jr, R. A. (2012). Orographic effects on precipitating clouds. Reviews of Geophysics, 50(1).

Trapero, L., et al (2013). Numerical modelling of heavy precipitation events over Eastern Pyrenees: Analysis of orographic effects. Atmospheric Research, 123, 368-383.

WMO (2008). Guide to meteorological instruments and methods of observation. WMO-No. 8. Seventh edition 2008, World Meteorological Organization, 681 pp, CH-1211 Geneva 2, Switzerland.

**Authors reply in bold.**

Referee #2

The authors provide a detailed account of the extreme weather events that occurred during June 1925 in south-western Spain using newspaper reports, station data and reanalysis data. The topic is of high-importance and the events are 1ocalized1 very well, though I think the meteorological analysis is limited and could be strengthened. Various methods are also not explained particularly well.

**Thank you very much for your comments. We believe that these extreme weather events that occurred during June 1925 in south-western Spain deserve the attention of the international scientific community. We will try to improve our meteorological analysis and we will also try to explain our methods better.**

My first major comment is regarding the choice of variables to analyse. The authors do not justify why they decided to look at SLP, CAPE and Precipitable Water. Since the events seem to be largely convective in nature the analysis would be strengthened by looking at other convective parameters such as vertical wind shear. Vertical velocity would also give an idea of the lifting available.

**The synoptic analysis of this situation was carried out with several variables. In our description we use SLP (because of its basic nature), CAPE (because the reports describe important thunderstorms) and Precipitable water (since many of the impacts are related to precipitation).**

**In any case, we have reviewed our results with the vertical wind and other variables. These results corroborate the results already described in the manuscript and, in order not to increase its length further, they have not been included in it.**

**As a non-exhaustive sample, the following figures show daily averages of v-wind from 20th Century Reanalysis v3 for Jun 8, 1925. This variable reveals the areas where increases (negative values) or decreases (positive values) are occurring. Simultaneous use on several levels can give an idea of the depth of the vertical ascents from the reanalysis point of view. In the proposed window, pressure levels of 925, 850, 700 and 500 hPa are shown.**

**As one can see, the southwest region of IP present negative values of v-wind at all the selected levels, indicating upward movements of air masses to very high levels of the atmosphere. These movements are the result of instability in the study area that could cause the existence of the case study thunderstorms.**

[Figure]

My second major comment is regarding the importance of the results and putting this in the context of previous literature. In much of section 5 the authors simply state values of certain variables and don't discuss why/how this favoured the development of the extreme events. The authors also make little to no comparison of how their results compare to previous literature.

**Thank you very much for your comment. We have reviewed the text of section 5 trying to put everything in the context of the previous literature as suggested by the referee. We have reviewed the existing literature to put our work in context.**

I am also not sure about the suitability of the journal. NHESS states "2ocalized case studies with no broader implications" are generally considered out-of-scope. The authors do not discuss the broader applications of their work in the current manuscript. I would recommend the authors strengthen such aspects if they wish to publish in this journal.

**Thanks for this comment. Although we are convinced that NHESS is an appropriate journal for our manuscript, we have taken your comment especially into account.**

Inline comments

L1–2. As far as I can see the authors only use Spanish newspaper reports for the analysis so wouldn't it make more sense for the title to be SW Spain rather than SW Iberia? Furthermore,

only station data for Spain is shown. SW Iberia presumably includes parts of Portugal which is not mentioned at all in the manuscript.

**We agree that we can change "Iberia" in the title and write "Spain", since most of the impacts detected were in Spain.**

L13–15. This sentence is overly long and the end of sentence on L15 does not read well. I'd recommend rewriting it in the following way "…..due to the large number of thunderstorms associated with significant loss to human lives and material resources".

**Thank you very much. We have changed this sentence. The new version is: "In a routine search for meteorological events with a great impact on society in the Extremadura region (SW interior of Iberia) using newspapers, the month of June 1925 was detected as exceptional due to the large number of thunderstorms associated with significant loss of human lives and material resources."**

L14. I would suggest thunderstorms in place of electrical storms throughout the manuscript.

**We agree. We have made that change throughout the entire manuscript. We have written "thunderstorm" instead of "electrical storm" in all cases.**

L16–23. The rest of the abstract is just one sentence which should be broken down for readability and clarity. The current abstract only has two sentences.

**We have re-written these lines. The new text is the following: "This extraordinary month underwent a detailed examination from various, complementary perspectives. Firstly, we reconstructed the history of the events, considering the most impacted locations and the resulting damages. Periodical publications, especially the widely circulated "Extremadura" newspaper in 1925, were pivotal in this regard. Secondly, we scrutinized monthly meteorological variables (precipitation, temperature, and cloudiness) using the lengthiest available data series in Iberia. This aimed to underscore the exceptional characteristics of June 1925. Lastly, we analyzed the synoptic situation of the thunderstorm events by employing 20CR reanalysis data. This approach allowed us to comprehend, from a synoptic perspective, the exceptional nature of this month. Thereby, a combination of a negative North Atlantic Oscillation (NAO) situation, elevated Convective Available Potential Energy (CAPE) values, and abundant total water vapor availability in the region was revealed."**

Why is abstract mostly focused on the methods and not the key findings?

**We have modified the wording of the abstract to change our approach. Please see the new version of the abstract in the previous answers.**

L23: "available water" Are authors referring to precipitable water?

**Yes, we are referring to precipitable water obtained from 20CR v3 reanalysis. This will be clarified in the revised version of the manuscript.**

L25–50: The authors cite several studies but only mention that these studies looked at various aspects of convection or thunderstorms. I would like to see more discussion in the introduction regarding the findings of these studies.

**We have modified section 1 of our manuscript according to the referee comments. In any case, our idea was to contextualize our work within the framework of thunderstorm studies in Iberia, but not to carry out an exhaustive review on this topic.**

L79: It is not clear to me how cloudiness is defined. I'd recommend writing a line saying something like: "Cloudiness is defined as….."

**According to the reviewer's suggestion, we have clarified this issue in the revised version of the manuscript. Thus, the parameter of cloudiness (PC) used in our work to characterize the cloudiness is defined (in percentage) as:**

**PC = 50 + 50 · ((O – C)/N)    (1)**

**where O and C are the number of overcast and cloudless days, respectively, and N is the number of days in a given period (month, season, year).**

**We have used the data provided by Sánchez Lorenzo et al. (2012) who inferred monthly series of the variable given by equation 1 from the number of cloudless and overcast days recorded every month in 39 Spanish stations since 1866. For that, those authors recovered monthly series of cloudless and overcast days since 1865 from different volumes of the publications entitled "Resumen de las observaciones meteorológicas efectuadas en la Península", edited by AEMET, from 1865 to 1950.**

**Reference:**

**Sanchez-Lorenzo, A., Calbó, J., and Wild, M.: Increasing cloud cover in the 20th century: review and new findings in Spain. 374 Clim. Past, 8, 1199–1212, doi:10.5194/cp-8-1199-2012, 2012.**

L86–90: The NOAA/CIRES/DOE 20th Century Reanalysis will not be familiar to all readers. The methods used to reconstruct atmospheric variables in this dataset should be briefly mentioned in the methods section. Additionally, how much can we rely on the data back in 1925? Limitations of this dataset are not discussed at all in the manuscript.

**We agree with this comment. We have included in the revised version of the manuscript additional information about the methods used in the 20CR reanalysis as well as its limitations, especially in the upper layers of the atmosphere. Evaluating the performance of the 20CR reanalysis in the historical part is not a simple task since it is impossible to make comparisons with other reanalyses and can only be done by comparison with independent observations (Slivinski et al., 2021). Some comparison exercises carried out have been satisfactory. In particular, in our study area, the 20CR results were satisfactory for the extreme precipitation event of autumn 1876 in the Guadiana River basin (Trigo et al., 2014).**

**Slivinski et al. (2021) An Evaluation of the Performance of the Twentieth Century Reanalysis Version 3. Journal of Climate 34, 1417. https://doi.org/10.1175/JCLI-D-20-0505.1**

**Trigo et al. (2014) The record precipitation and flood event in Iberia in December 1876: description and synoptic analysis. Front. Earth Sci. 2, 3. doi: 10.3389/feart.2014.00003**

Which version of the dataset did the authors use? Version 3 covers 1836–2015 but the authors say the variables that they had were available dating back to 1871 which is the availability for version 2. The spatial and temporal resolution of the dataset should also be added.

**Thank you for your comment. We have used version 3. We have clarified this and have also incorporated some details about the spatial and temporal resolution of this dataset in the new version of the manuscript.**

L108: "many lightning struck" sounds a bit unnatural. I'd suggest writing "during which there were several lightning strikes, one of which…"

**Thank you very much for this style suggestion. We accept this change.**

L109: Generalized is not usually used in this context in English. I think the authors mean to say "a widespread power blackout".

**Thank you very much again for this style suggestion. We accept this change.**

L110: How large were the hailstones? What was the nature of the damage in the countryside? I think it would be useful to add this information.

**We have tried to locate some information about the size of the hail in the news reports published in the newspapers. However, these efforts have been null. Newspaper reports only indicate (sometimes) "hail", without specifying the size, so we assume that the hail was not large. The main damages mentioned in the recovered news were caused by flooding and lightning.**

L124: I'd suggest using "the fatalities" instead of "these dead people". "Dead people" sounds a bit too harsh for a scientific text. The sentence would read better if it were written as "As well as the fatalities, there were several injured people and deceased animals."

**Thank you very much again for this style suggestion. We accept this change.**

General comment on section 3: I think this section could be shortened. For example, the number of each animal which died in each region is mentioned and sometimes how they died is described. I don't think such specific information is relevant.

**We have shortened the length of section 3 in the next version of the manuscript, eliminating information of little interest such as that cited by referee #2. In particular, we have eliminated the following text, as well as other minor deletions:**

**"…because they received an electric shock when they stumbled into a telephone cable that had come off."**

**"In Cáceres, twelve hens and six sheeps disappeared by the water. In Zafra, the overflowing of the river Bodión swept away animals on June 10th, which also happened in Montemolín when the streams overflowed, according to the reports of the newspaper "Extremadura". Furthermore, many animals also perished due to lightning strikes. That was the case of fifty one hens and one donkey in Segura de León.**

L134: A flood is usually due to overflowing water so the "overflowings" part here is redundant. Overflowing in the plural form does not exist in English.

**Thank you very much again for this style suggestion. We accept this change.**

L159: Can the authors also add a map of Spain with the Extremadura region highlighted? It may also be nice to add some topographic features.

**Figure 2 is showing a map of Iberia with the borders of the region of Extremadura (and its two provinces) including topographic features. In any case, we will added more information (in text and figures) about this issue as we have indicated in our responses to referee #1.**

L169: 20 mm day-1

**Thank you very much again for this style suggestion. We accept this change.**

L183: Why did the authors standardize the anomalies? Why not just show the anomalies in kelvin (temperature) and millimetres (precipitation)? It is also still not clear to me how cloudiness is measured in this study.

**The standardized anomalies are calculated as the differences between June 1925 and the whole period, and then scaled by the division of the whole period standard deviation. They generally provide more information about the magnitude of the anomalies because influences of dispersion and location have been removed from data. Thus, the standardized anomalies measure an average departure from the mean in terms of the number of standard deviations.**

**We have used standardized anomalies because the three meteorological variables (temperature, precipitation and cloudiness) exhibit clear seasonal variations. Thus, standardized anomalies provide more information about the magnitude of the anomalies because influences of dispersion have been removed.**

**Cloudiness is derived in our study from equation 1 (see above) using the number of cloudless and overcast days recorded every month in 39 Spanish stations since 1866.**

L201–202: Thunderstorms usually occur after a prolonged warm spell of weather, so this statement confuses me a bit. It seems that a cut-off low pressure system was a prominent pattern during June 1925, with perhaps then embedded convection enhancing the rainfall locally. This could explain the increased cloudiness and lower temperatures.

**This is not usual in Extremadura. In this region, thunderstorms are normally produced by an increase in instability of dynamic origin, an advection of vorticity due to the arrival of a front, or another mechanism. Storms with thermal origin only occur in summer and are not that frequent.**

L206–207: A clear dependence on latitude can be seen, with negative cloudiness anomalies for all northern locations and positive anomalies for the central and southern sites.

**Thank you very much again for this style suggestion. We accept this change.**

L222–223. Can a marker be added to each figure where a storm occurred, so it is easier to identify which synoptic regimes were associated with storms?

**We fear that the realization of this idea is not entirely possible. On this synoptic scale, the Extremadura region occupies very few pixels in the figure. We perfectly understand the idea suggested by the referee, so what we have done is to indicate the outline of the Extremadura region so that readers have a graphic reference of where the thunderstorms occurred.**

It seems a cut-off low persisted from around June 3$^{rd}$–June 8$^{th}$. The authors do mention the cut-off low but do not discuss whether this was a contributing factor to the extreme events.

**Yes, we agree. Thank you so much. The cut-off low pressure system was one of the prominent patterns during June 1925. We are convinced that the corresponding convection increased precipitation that was very intense locally. This could also explain the increase in cloudiness and lower temperatures than usual for the month of June in this region. We have expanded on what we have written about cut-off low systems in the corresponding section of the new manuscript (second pharagraph of section 5).**

L240–243: Where is all this information coming from? Did Santos et al. 2019 show which patterns are typically associated with which weather? I think an extra sentence clarifying this would be useful.

**We believe that we have not been able to explain our work well. Indeed, the Spanish Meteorological Agency (AEMET) published an update of the synoptic classification usually used by this agency (Santos et al. 2015). In this study by Santos et al. (2015), using the ERA40 reanalyses, the objective classification of Ribalaygua-Batalla and Borén-Iglesias (1995) is reviewed, and the subjective classification of Font (1983) is recovered in detail, which proposes 23 synoptic patterns, illustrated with situations of 23 specific dates, in general from the 1970s-1980s. We will add some sentences clarifying these lines 240-243. In addition, the document by Santos et al (2015) is freely accessible at the following web address:**

**https://www.aemet.es/es/conocermas/recursos_en_linea/publicaciones_y_estudios/publicaciones/detalles/NT_27_AEMET**

**Font-Tullot, I., 1983. Climatología de España y Portugal. Instituto Nacionalde Meteorología, 1983. Madrid.**

**Ribalaygua-Batalla, J., Borén-Iglesias, R., 1995. Clasificación de patrones espaciales de precipitación diaria sobre la España peninsular y Baleárica. Informe Nº 3 del Servicio de Análisis e Investigación del Clima. INM. Madrid.**

L245–246: I am not sure what the authors want to say here. It reads as if the newspapers carried out a synoptic analysis which is consistent with the authors' synoptic analysis.

**Probably, we have not explained this well. We have rewritten the sentences this way: "As evident from Section 3 and Figure 4, most stormy and rainy days occurred from day 1 to 22. Consequently, the synoptic analysis conducted in this section aligns with the observations documented in the newspapers."**

L257–258. How do you conclude these are high CAPE values? Can you provide any reference values for what is considered a high monthly mean of CAPE?

**Thank you very much for this comment. Indeed, we believe that some additional explanation is necessary to explain the CAPE values shown. Any meteorology book that explains how to calculate CAPE values from an aerological diagram indicates that we have an extremely unstable atmosphere for CAPE values greater than 3500 J/kg. The values shown in Figure 9 present maximum CAPE values of the order of 150 J/kg. This may be surprising to some readers. However, one must keep in mind**

**that the values shown correspond to the composite mean of the entire month. Therefore, it is correct that these apparently low values appear. Normal values for the climate of Extremadura are below 50 J/kg (composite mean). We will add some sentences to better explain these values.**

Additionally, Figure 9 shows the largest CAPE in Spain for June 1925 was in north-western Spain and northern Portugal, away from the region with the highest precipitation anomalies in south-western Spain. CAPE in south-western Spain was relatively low in comparison. Does this mean that CAPE was not the primary driver of the extreme events in the far south-west? It is worth keeping in mind that high CAPE is not necessarily a prerequisite for extreme precipitation events and flooding, especially if a cut-off low lingers for several days.

**We are aware of these details that could be inconsistent. We believe that these details are not very relevant in this case because the 20CR reanalysis for such early times gives us significant patterns although perhaps the exact location of the details is a little displaced. We have incorporated some sentences to explain this in the main text.**

L258: How was this anomaly calculated, is it a monthly anomaly? When is the reference period?

**This monthly anomaly is calculated from the composite mean value. Please see also our extensive response to your comments on lines 257–258. Details about the calculus and plots from 20CR can be found in https://psl.noaa.gov/data/composites/20thc_rean/details.html.**

**Plots are of mean-climatology for each month. Climatology time period selected for the calculus is 1981-2010. We will add some sentences explaining these details.**

L265–266: Why did the authors chose to calculate monthly means? It seems the extreme weather reports are available on a daily scale so why not get a better understanding of the environments in which individual storms formed by looking at daily or sub-daily data?

**We are also calculated and plotted maps with daily and subdaily values, but values are similar to monthly ones (bridging the timescale differences) and we decided not show them. Moreover, daily information is included and documented in Figures 7 and 8, that reveal really anomalous situations with persistence during most of the days of the month June of 1925 year.**

L310–311: low pressure

**Thank you very much again for this style suggestion. We accept this change.**

---

## Author Response (AR2)

******************Editor******************

Dear Authors,

I am glad to inform you that your paper can be accepted subject to the minor corrections proposed by the reviewer. Please correct the manuscript accordingly. Thank you for considering NHESS for the publication of your research.

Sincerely

V. Kotroni

Editor

**Thank you very much for your message. We have corrected our manuscript according to the corrections proposed by both reviewers. We hope that this new version of our manuscript can be accepted for publication in NHESS.**

******************Referee 1******************

1. Page 3, line 71. Table 1 title. Suggest: "Date, newspaper name, title, …" -> "Date, newspaper name, title [translated title], …"

**Thank you. We have accepted this suggestion.**

2. Page 3, Table 1, typo: Tittle

**Thank you. We have corrected this typo in Table 1.**

3. Page 3, Table 1, 2nd column, third row (item), typo: cadaver (accent missing on the second 'a')

**Thank you. We have corrected this typo in Table 1.**

4. Page 4, line 124, typo: norther

**Thank you. We have corrected this typo (page 6, line 124).**

5. Page 5, lines 101-102: This sentence has no verb; what about combining it with the previous one: "… data assimilation method, thus providing a direct estimation of the most likely state of the global atmosphere (for each three-hour period)."?

**Thank you very much. We have corrected this sentence without verb combining it with the previous one, as suggested by the referee.**

6. Page 5, line 102: there also is an -> there is also an

**Thank you. We have corrected the order of these words.**

7. Page 5, last paragraph. Authors should make clear that the upper-level information (250 hPa) from the reanalysis used for the period examined was derived essentially using statistical methods and is not the result of a standard reanalysis. This means it has a level of uncertainty much higher than sea-level pressure fields or upper-level information for periods where radiosonde information is available. This remark is important as several statements are made

using 250 hPa level data which should be taken with caution – the current text seems they correspond to a standard reanalysis.

**Thank you for this comment. We have added to this paragraph the following sentences: "The upper level (250 hPa) information from the 20CR reanalysis will also be used in this work. It should be noted that it was derived primarily by statistical methods for the period examined and is not the result of a standard reanalysis. This means that it has a much higher level of uncertainty than the sea level pressure fields or the upper level information for periods where radiosonde information is available."**

8. Page 6, 1st paragraph. Some locations mentioned in the text are not shown in Figure 3 such as river Tajo or the "Sierra de Gata and Sierra de San Pedro" – please make sure that all cited locations are shown so that readers can follow the text.

**We have included in the new version of the manuscript a new Figure 3 incorporating the suggestions made by the referee.**

9. Page 8, Figure 3. The names of the rivers can hardly be read; please modify the font used to make them clearer.

**We have included in the new version of the manuscript a new Figure 3 incorporating the suggestions made by the referee.**

10. Page 8, Figure 3 caption, line 183. Typo: means -> mean

**Thank you. We have corrected this typo (page 8, line 183).**

11. Page 9, Figure 4. Could you please indicate in the map (Figure 3) the locations of the 7 rain gauge stations shown in Figure 4?

**Please, note that the locations of the 7 rain gauge stations shown in Figure 4 are indicated in Figure 2.**

12. Page 10, Equation 2 (and line 205). There's still some problem with the symbol used to indicate the average value – please correct.

**We are sorry. It has already been corrected.**

13. Page 10, line 208, typo: times series should be time series.

**Thank you. We have corrected this typo (page 10, line 208).**

14. Page 10, Figure 5 (and Figure 6). Please add the degree units to the axis titles.

**We have included new versions of Figures 5 and 6 adding the degree units to the axis titles.**

15. Page 17, Figure 9. Please remove the vertical and horizontal lines near the borders.

**Thank you. This problem was solved.**

******************Referee 2*******************

The authors have addressed the majority of my comments, especially the inline comments. The methodology is now much clearer than the original version. The synoptic analysis (section 5) is still limited and could be expanded by looking at daily data and including more variables, but this could also be a focus of future work. Unfortunately, there is still little comparison to previous literature. I recommend further revisions before publishing.

Section 5

• The authors decide to not include any further meteorological variables and stick with sea level pressure, precipitable water content and CAPE. The limitation to this is that thunderstorms and deep moist convection require three ingredients: moisture, instability and lift (Doswell et al. 1996). Vertical wind shear is also required to allow storm organisation (e.g. Markowski and Richardson, 2010). In the current manuscript, the authors only look at the first two ingredients: precipitable water content (moisture) and CAPE (instability). Sufficient instability and moisture are not enough to allow thunderstorm initiation, lifting (a triggering mechanism) is also required. Whilst there are numerous sources of lift including very local sources, I think that at a minimum the vertical velocity anomaly in the reanalysis should be included. Indeed, there is a negative anomaly in Omega at 500 hPa during June 1925 in western Iberia (NOAA Physical Sciences Laboratory, 2024). While the sea level pressure anomaly is negative across much of Europe (Figure 9), western Iberia stands out as a region with stronger large-scale lifting than average when looking at the Omega field. This indicates that large-scale lifting could have been a relevant factor for the development of the thunderstorms in June 1925.

•Some nice discussion has been added in section 5, especially about the links between the cut-off low and how this would have affected the moisture advection. However, comparison to previous literature is still lacking. The only citations in section 5 are pointing to the classification scheme that they used.

**Thank you very much for this comment and suggestion. We agree. lifting (a triggering mechanism) is also required. Therefore, we have re-written a large part of Section 5 and we have included a new Figure 10. The new material is copied below.**

**Lastly, we have generated synoptic charts of the main meteorological fields, as well as different composites of the monthly mean values and anomalies regarding the climatological period covered by the 20CR reanalysis. Following Doswell et al. (1996), thunderstorms and deep moist convection require three ingredients: moisture, instability and lift. Vertical wind shear is also required to allow storm organization (e.g. Markowski and Richardson, 2010). In the current manuscript, precipitable water content (moisture), CAPE (instability) and Omega (dp/dt, lifting) are analyzed.**

**A summary of our results is presented in Figure 9 and Figure 10. Figure 9 is made up of six panels. The top two panels show SLP while the middle two panels depict Convective Available Potential Energy (CAPE) and the bottom two panels display total precipitable water. The panels on the right present the composite means of the variables indicated for June 1925 while the panels on the left exhibit the composite anomaly.**

**The top panels of Figure 9 show a typical negative North Atlantic Oscillation (NAO) situation with low pressures west of the British Isles and negative SLP anomalies in**

southwestern Iberia. The middle panels of Figure 9 reveal that western Iberia had high CAPE values in the context of the Atlantic and Mediterranean region, with positive mean anomalies in western Iberia during June 1925 (the values shown correspond to the composite mean of the entire month). Finally, the bottom panels present high values of precipitable water in the entire atmosphere in southwestern Iberia with the highest values of the anomaly over the region of Extremadura. Note that these monthly anomalies are calculated from the composite mean value (climatology time period selected for the calculation is 1981-2010). Therefore, the exceptional month of June 1925 in Extremadura was characterized by a combination of negative NAO situation, high CAPE values, and total water vapor available in this area. In any case, note that Figure 9 shows the largest CAPE in Spain for June 1925 was not located exactly in the south-western Spain but in north-western Spain and northern Portugal. It seems the 20CR reanalysis for such early times gives us significant patterns although perhaps the exact location of the details is a little displaced.

Panels in Figure 9 are complemented by panels in Figure 10, that show Omega field at several pressure levels during June 1925 (left panels) as well as Omega field at the same pressure levels during all of June months for the period 1851-2014 except 1925 (right panels). Results on left panels show a negative anomaly in Omega at all pressure levels during June 1925 in western Iberia until a very high level far from surface (~150 hPa) where the lift seems to have disappeared. Moreover, while the sea level pressure anomaly is negative across much of Europe (Figure 9), western Iberia stands out as a region with stronger large-scale lifting than average when looking at the Omega field. This indicates that large-scale lifting could have been a relevant factor for the development of the thunderstorms in June 1925. Results on right panels show this negative anomaly is non-existent for June months for the full period of reanalysis 1851-2014 except 1925. It indicates the exceptionality of the month June 1925 treated on this work and not included in the ESWD (Dotzek et al., 2009) which only includes four severe weather phenomena for the year 1925 in eastern Spain.

**Added references:**

Doswell, C. A., Brooks, H. E., and Maddox, R. A.: Flash Flood Forecasting: An Ingredients-Based Methodology, Weather Forecast., 11, 560–581, https://doi.org/10.1175/1520-0434(1996)011<0560:FFFAIB>2.0.CO;2, 1996

Dotzek, N., P. Groenemeijer, B. Feuerstein, and A. Holzer, 2009: Overview of ESSL's severe convective storms research using the European Severe Weather Database ESWD. Atmos. Res., 93, 575–586, https://doi.org/10.1016/j.atmosres.2008.10.020.

Markowski, P. and Richardson, Y.: Mesoscale Meteorology in Midlatitudes, in: vol. 2 of Advancing weather and climate science, 1st Edn., Wiley, Somerset, ISBN 0470742135, 2010.

[Figure]

**Figure 10: Composite anomaly for June 1925 (left panels) and composite anomaly for June months from 1851 to 2014 except 1925(right panels) of Omega (dp/dt) in the study area for pressure levels of 600, 500, 400 200 and 150 hPa(from top to bottom panels, respectively) from 20CR Reanalysis.**

Inline comments:

L23: The authors now talk about "total water vapor availability". In Figure 9 precipitable water is analysed. Total column water vapour is a separate variable so this could be misleading. I'd recommend using consistent terminology.

**We agree with this point. We have changed line 23 from "total water vapor availability" to "precipitable water".**

L28–29: The authors should directly reference the European Severe Weather Database (ESWD; Dotzek et al. 2009), which is the main convective storm database used in Europe.

**Thank you very much. We have referenced this valuable work by Dotzek et al. (2009). We mentioned the European Severe Weather Database (ESWD; Dotzek et al. 2009) in sections 1 and 5.**

L87: I think it would be useful to add a line on how Sánchez Lorenzo et al. (2012) defined an overcast day and cloudless day. If it is cloudy for 30 minutes of the day is this considered an overcast day? Must it be sunny for the entire day for it to be considered a cloudless day?

**As is noted in our work, Sanchez-Lorenzo et al. (2012) recovered monthly series of cloudless and overcast days since 1865 from different volumes of the publications entitled "Resumen de las observaciones meteorológicas efectuadas en la Península", edited by Spanish Weather Service from 1865 to 1950. According to these authors, it was not possible to find the original human observations of daily cloud cover (most likely in oktas or tenths) made at the 39 meteorological stations analyzed in their work. Thus, it is not possible to response the questions reported by the reviewer because the information is not known. We have tried to partly clarify this issue, adding the following sentences to the revised version of the manuscript:**

**"Capel Molina (1981) established that a day is defined as cloudless if the mean cloud cover from several daily observations is lower than 20%, while is defined as overcast if this mean is higher than 80%. Thus, if the cloud cover is recorded in oktas the thresholds could be less than 1.5 for cloudless days and greater than 6.5 for overcast days."**

**Capel Molina, J.J.: Los climas de España. Ed. Oikos-tau, 1981.**

L102–106: I appreciate that the authors have included further details on the 20th Century Reanalysis. If the observations shown in Figure 6 are independent, then the authors could show a comparison between observations and the 20th Century Reanalysis in the Appendix. This would support the authors' claim that "some comparison exercises carried out have been satisfactory".

**We understand this point suggested by the referee. However, a direct comparison between observations and the 20th Century Reanalysis is not a simple task. Moreover, we are sure that a part of the observations shown in Figure 6 are not independent. In any case, we have made some simple approximations as a test. In them, we see that a reasonable likelihood between the observed data and the modeled data is verified. In particular, we can highlight that the general patterns of the meteorological fields are reproduced (although they may be slightly displaced in their location). As an example, we present a plot of the temperature anomaly in June 1925 obtained with the 20CR reanalysis. It can be clearly seen in this plot that the**

**general pattern is similar to that obtained with the observational data we present in our manuscript with negative anomalies in the SW of Spain.**

[Figure]

L302: calculus calculations. Calculus involves differential equations.

**Thank you. We have changed this word.**

L303: "total water vapour available". See comment above on L23.

**Thank you very much. We have changed line 23 from "total water vapor availability" to "precipitable water".**